# Endogenous TOM20 Proximity Labeling: A Swiss-Knife for the Study of Mitochondrial Proteins in Human Cells

**DOI:** 10.3390/ijms24119604

**Published:** 2023-05-31

**Authors:** Sébastien Meurant, Lorris Mauclet, Marc Dieu, Thierry Arnould, Sven Eyckerman, Patricia Renard

**Affiliations:** 1URBC, Namur Research Institute for Life Sciences (Narilis), University of Namur (UNamur), 5000 Namur, Belgium; sebastien.meurant@unamur.be (S.M.);; 2Mass Spectrometry Platform (MaSUN), Namur Research Institute for Life Sciences (Narilis), University of Namur (UNamur), 5000 Namur, Belgium; 3VIB-UGent Center for Medical Biotechnology, VIB, 9000 Ghent, Belgium; 4Department of Biomolecular Medicine, Ghent University, 9000 Ghent, Belgium

**Keywords:** mitochondria, co-translational import, BioID, protein identification, mass spectrometry

## Abstract

Biotin-based proximity labeling approaches, such as BioID, have demonstrated their use for the study of mitochondria proteomes in living cells. The use of genetically engineered BioID cell lines enables the detailed characterization of poorly characterized processes such as mitochondrial co-translational import. In this process, translation is coupled to the translocation of the mitochondrial proteins, alleviating the energy cost typically associated with the post-translational import relying on chaperone systems. However, the mechanisms are still unclear with only few actors identified but none that have been described in mammals yet. We thus profiled the TOM20 proxisome using BioID, assuming that some of the identified proteins could be molecular actors of the co-translational import in human cells. The obtained results showed a high enrichment of RNA binding proteins close to the TOM complex. However, for the few selected candidates, we could not demonstrate a role in the mitochondrial co-translational import process. Nonetheless, we were able to demonstrate additional uses of our BioID cell line. Indeed, the experimental approach used in this study is thus proposed for the identification of mitochondrial co-translational import effectors and for the monitoring of protein entry inside mitochondria with a potential application in the prediction of mitochondrial protein half-life.

## 1. Introduction

Biotin-based proximity labeling has gained increasing interest in the last decades and has proven to be a useful tool for the unbiased in vivo discovery of vicinal proteins [1,2,3,4] and the identification of organellar proteomes [5,6,7,8,9]. The main advantage of this technique, when compared to other approaches of proximity labeling, is the use of living cells and the fact that it enables the identification of both direct protein partners and of proteins in close vicinity of the bait protein [10,11,12]. In addition, some biotin labeling variants such as APEX2 support very brief labeling times, providing true snapshots of the protein complexes, while the classical BirA*-based approach requires longer labeling times, also revealing weaker associations [11,13]. However, suitable controls and careful validations are required to discriminate direct partners from non-interacting proteins in the vicinity of the bait protein and to exclude background proteins [11]. Both complete mitochondrial proteomes and submitochondrial proteomes have been characterized using this technology, taking advantage of the closed and compartmentalized feature of the organelle [5,14,15]. Moreover, the biotin-based proximity labeling approach (also called BioID) has been demonstrated to be a useful tool for the characterization of the mitochondrial surface, not only in terms of proteins [8,9,14] but also in terms of transcripts [7,16,17,18]. The generation and development of split versions of the biotin ligases also enable to go further and to identify the specific factors of inter-organelle contact sites with, for example, the identification of mitochondria-associated membranes (MAMs) proteome [8,9]. There is great interest in the use of protein-fused or membrane-anchored APEX peroxidases for the mapping of the whole mitochondrial proteome and the proteome of the different submitochondrial compartments, identifying altogether up to two-thirds of the mitochondrial proteome [5,14,15]. The advantage of this enzyme is its high labeling efficiency with the generation of short living biotin-phenol radicals (~1 ms), leading to an average labeling radius of <20 nm and giving snapshots of the proxisome [5,13]. Due to the very low permeability of the radicals, the use of APEX enzymes targeted to closed compartments, such as the mitochondrial matrix, was revealed to be useful for mitochondrial matrix proteome identification and for resolving membrane topology [5,19,20]. However, this technique requires a pulse of hydrogen peroxide, whereas the use of other biotinylating enzymes, such as the modified bacterial biotin ligase BirA* and its derivatives, does not require such treatment [11,13]. Additionally, those enzymes have a labeling radius estimated at 10 nm with a longer labeling time, requiring more than 16 h of labeling [10,11,13]. However, variants with a higher labeling efficiency were recently developed, lowering the labeling time window to 10 min of biotinylation [21,22].

The biotin-based proximity labeling assay has thus been extensively used to characterize the mitochondrial proteome, but this tool could be further applied for the study of mitochondrial protein import in living cells, with this process being critical for mitochondrial physiology [23]. Indeed, mitochondria are crucial and special organelles, and although they possess their own genome, which encodes 13 peptides of the electron transport chain, mitochondria import the bulk of their proteins (about 1500 different proteins in human cells) from the cytosolic compartment [24]. Several mitochondrial protein import pathways have been described [25,26,27], with the so-called pre-sequence pathway, responsible for the transport of 60% of mitochondrial proteins in the matrix, being the most well characterized. Briefly, it transports precursor proteins containing an N-terminal mitochondrial targeting sequence (MTS) through the translocase of the outer membrane (TOM) and of the inner membrane (TIM23) complexes. The membrane potential of the inner membrane drives the translocation of the positively charged MTS, which is finally cleaved by the matrix mitochondrial processing peptidase [25]. The MTS are typically recognized by the TOM receptors, TOM20 and TOM22, embedded in the outer mitochondrial membrane (OMM), and then translocated through the TOM40 channel. However, the TOM40 channel is quite narrow and only allows the translocation of unfolded or loosely folded proteins [28,29]. This leaves only two possibilities to import nuclear-encoded proteins into the mitochondria: either a post-translational, or a co-translational import mechanism. In the post-translational import, the proteins synthesized by cytosolic ribosomes are targeted to mitochondria by chaperones that maintain them in an unfolded state, followed by an import through the TOM complex. In the co-translational import mechanism, nascent polypeptides are directly imported through the TOM40 channel concomitantly with their synthesis by the ribosome localized at the surface of the OMM [23,30].

The post-translational import pathway is largely described but this chaperone-dependent system is energy costly and prone to the aggregation of hydrophobic proteins in the hydrophilic environment of the cytosol [23,25]. This supports the observations that a co-translational import mechanism co-exists, at least in yeast and *Drosophila*, enabling the avoidance of energy expenditure by the chaperone and co-chaperone systems [27,31,32,33]. Several molecular actors of this process have been identified and characterized in yeast, such as the mitochondrial receptor of the ribosome Outer Membrane 14 protein (OM14) [33,34] or the Puf3 RNA binding proteins [35,36]. OM14 interacts with the nascent chain-associated complex (NAC), a dimeric complex associated with the ribosome exit site and involved in the transport of the nascent peptide to specific subcellular locations such as the mitochondria. Thus, OM14 acts as a mitochondrial receptor for the NAC-associated nucleus-encoded mitochondrial transcripts and promotes the interaction between the nascent peptide, TOM20 and the import machinery upon translation [34]. Puf3, a member of the Pumilio RNA-binding protein family, binds specific sequences in the 3′UTR of several nucleus-encoded mitochondrial transcripts and induces their OMM localized translation and co-translational import by favoring the interaction between the nascent peptide and TOM20 [37,38,39]. In *Drosophila*, three proteins have been shown to promote the localized translation of nucleus-encoded mitochondrial transcripts at the surface of mitochondria, facilitating the mitochondrial co-translational import. First, the Phosphatase and Tensin homolog (PTEN)-induced putative kinase 1 (PINK1) associates with both specific transcripts encoding mitochondrial proteins and the translation machinery and represses the translational repressor hnRNP/Glo at the surface of the mitochondria, promoting the translation of nucleus-encoded mitochondrial transcripts close to TOM20 [40]. Alternatively, under conditions of mutated mitochondrial DNA, PINK1 can also negatively regulate other translational regulators, inhibiting the localized translation of nucleus-encoded mitochondrial transcripts at the surface of mitochondria to safeguard mitochondrial genome integrity [41]. Second, MDI (mitochondrial DNA insufficient), the *Drosophila* ortholog of the scaffold protein AKAP1, recruits the La-related RNA binding protein (Larp) to the mitochondrial surface, promoting the translation of nucleus-encoded mitochondrial transcripts encoding replication factors, mitochondrial ribosomal proteins and electron transport chain (ETC) proteins [42]. Third, Clueless (Clu) protein, located at the OMM, associates with PINK1-Parkin [43], and binds both mitochondria transcripts and ribosomes, and is therefore proposed to mediate mitochondrial co-translational import [44]. The above-cited examples represent the three main categories of proteins mediating or favoring mitochondrial co-translational import: (1) receptors for ribosomes at the surface of mitochondria; (2) RNA-binding proteins (RBPs) targeting nucleus-encoded mitochondrial transcripts at the surface of mitochondria; and (3) translational regulators anchored at the surface of mitochondria, promoting the translation of nucleus-encoded mitochondrial transcripts close to the TOM complex. However, in mammals, no protein has been specifically demonstrated to participate in the mitochondrial co-translational protein import, although evidence supports the existence of this process since some mitochondrial proteins have been demonstrated to be co-translationally imported in human cells [45]. Indeed, by using specific reporters, Mukhopadhyay and colleagues were able to discriminate between the post- and co-translational import of selected proteins and could describe the co-translational import of the Aldehyde Dehydrogenase 2 (ALDH2), the Arginase 2 (ARG2) and the Ornithine Transcarbamylase (OTC) [45]. In addition, nucleus-encoded mitochondrial transcripts have been shown to be enriched at the surface of mitochondria in different human cell lines [16,46,47]. Interestingly, a more recent study identified the transcripts undergoing translation in the vicinity of the TOM complex in human cells by taking advantage of the elegant proximity-specific ribosome profiling technique [18,48]. In this study, the authors highlighted the role of CLUstered mitochondria protein Homolog (CLUH), the mammalian ortholog of Clu, in the regulation of the mitochondrial localized translation of several nucleus-encoded mitochondrial transcripts, showing a potential involvement in the mitochondrial co-translational import process [18]. There are thus arguments that support the existence of such types of protein import in human cells, even if the effectors of this process are still unknown.

The utility of BioID approaches to characterize organelle surface/content does not need to be demonstrated more. In this study, we developed a TOM20-miniTurbo (TOM20-mTb) cell line to further characterize the proxisome of the endogenous TOM complex. With the use of specific BioID control, we tested this specific tool for the unbiased identification of mitochondrial co-translational import effectors. The first results revealed a high enrichment of RNA-binding proteins close to the TOM complex, supporting a co-translational import, or at least a localized translation of transcripts. However, by using a limited number of available co-translational reporters, we were not able to demonstrate the involvement of selected candidates in mitochondrial co-translational import. Interestingly, TOM20 proximity labeling highlighted new mitochondrially targeted nuclear proteins such as MED15, CPSF2 and GPATCH4, with no prior mitochondrial annotations. We additionally propose the use of the TOM20-mTb tool and experimental approach to evaluate mitochondrial protein half-life. Altogether, we present the TOM20-mTb construct as a useful tool both for studying mitochondrial co-translational import in mammals and for tracking protein entry inside mitochondria.

## 2. Results

### 2.1. Characterization of Mitochondrial Post- and Co-Translational Import Reporters

To assess mitochondrial co-translational import, specific reporters could be used but, unfortunately, robust reporters are lacking and have so far impaired the exhaustive study of this import pathway in humans. Indeed, very few mitochondrial proteins have been shown to be co-translationally imported into the mitochondria of human cells. Only three enzymes found in the matrix of the organelle have been shown to be specifically co-translationally imported: ALDH2, ARG2 and OTC [45]. Therefore, we constructed specific reporters by fusing the MTS + 10 amino acid residues of those enzymes to the N-terminal part of *E. coli* Dihydrofolate Reductase (ecDHFR), henceforth named preALDH2-DHFR, preARG2-DHFR and preOTC-DHFR reporters (Appendix A). Additionally, we also made a fourth construct using the pre-sequence of the Cytochrome c Oxidase complex subunit 4 (preCOX4I1), a nuclear-encoded subunit of complex IV of the electron transport chain (Appendix A). This complex is known to show a strong coordination for correct assembly between the two locations of the genes encoding the mitochondrial proteins of this complex and also between their translation and import [49,50]. In the transfected HCT116 cells expressing the reporters, all constructs display a mitochondrial localization as demonstrated by the colocalization of each construct with TOM20, a protein of the outer mitochondrial membrane (Figure 1A). The use of DHFR-based reporters was previously reported in studies aiming at describing mitochondrial co-translational import [45,51]. Indeed, in order to assess the co-translational import feature of each construct, we took advantage of the specificity of trimethoprim (TMP) to bind and stabilize the bacterial ecDHFR, preventing post-translational import of the construct [51,52]. Therefore, only a co-translationally imported construct will be detected inside the mitochondria in the presence of TMP. Unexpectedly, out of the three constructs previously reported to be co-translationally imported, only the preOTC-DHFR construct was fully imported into mitochondria upon trimethoprim treatment, whereas both preALDH2-DHFR and preARG2-DHFR were blocked outside the mitochondria (Figure 1B). An intermediary phenotype was observed for the preCOX4I1-DHFR construct with both mitochondrial and cytosolic localizations. PreALDH2-DHFR and preARG2-DHFR showed thus a post-translational feature and preCOX4I1-DHFR presented an intermediary phenotype with only preOTC-DHFR being fully co-translationally imported (Figure 1C). One explanation for the discrepancy observed for the preALDH2-DHFR and preARG2-DHFR reporters might be a cell-type specificity. Indeed, while we are studying HCT116 cells, the original study used HeLa cells [45]. Therefore, the reporter plasmids were also transfected in HeLa cells (Appendix A) as well as in HEK293T (Appendix A). A similar feature was observed for all four reporters. The import pathway guided by the four tested pre-sequences seems thus to be conserved between different cell types.

### 2.2. LARP4 Does Not Participate in the Co-Translational Import of preOTC-DHFR Reporter

Taking advantage of the preOTC-DHFR co-translational reporter, we focused on a couple of proteins susceptible to contribute to co-translational import: the scaffold protein A-Kinase Anchoring Protein-1 (AKAP1) and the translational regulator La Ribonucleoprotein 4 (LARP4). These two proteins are good candidates and likely mediate mitochondrial co-translational import in human cells. Indeed, MDI and Larp, their *Drosophila* orthologs, were previously demonstrated to ensure this function [41,42]. Moreover, these two proteins were recently shown to mediate the translation of transcripts encoding mitochondrial proteins at the surface of mitochondria in HEK293T cells [53] and thus seem to ensure the same function as MDI and Larp [42]. As previously shown [53], AKAP1 shows a mitochondrial localization (Figure 2A) whereas LARP4 seems to partially colocalize with the mitochondrial network in HEK293T cells (Figure 2B). A similar localization for both proteins was shown in normal human dermal fibroblasts (NHDFs) (Appendix A) and in HCT116 cells (Appendix A). The interaction between the two proteins was also confirmed in HCT116 cells by co-immunoprecipitation (Figure 2C). Therefore, to test the possible contribution of LARP4 to the mitochondrial co-translational import, we constructed LARP4 knock-out (KO) cell lines (Appendix A) and assessed the localization of the preOTC-DHFR co-translational reporter in those cells (Figure 2D). No modification of the mitochondrial localization for the protein encoded by the construct was observed in LARP4 KO cells, as confirmed by the high colocalization of the construct with the TOM20 marker (Figure 2E). However, as the reporter could be alternatively imported through the post-translational pathway, we first checked whether the post-translational import was affected by LARP4 KO. To do so, we used the post-translational reporter preALDH2-DHFR but no modification of its mitochondrial localization could be observed in LARP4 KO cells (Appendix A), confirming the integrity of this import pathway. Therefore, to test whether our co-translational reporter is alternatively post-translationally imported in this condition or not, we added trimethoprim to cells transfected with the preOTC-DHFR construct to block its putative post-translational import (Appendix A). Even upon trimethoprim treatment, the encoded reporter protein still localized inside the mitochondria in LARP4 KO cells (Appendix A). These results highlight the limitations of our single reporter, which is probably not sufficient to identify by itself the mitochondrial co-translational import effectors. Indeed, one can hypothesize that diverse proteins contribute to mitochondrial co-translational import via different mechanisms, in a pre-sequence-specific manner. Additional reporters are thus required to identify the effectors of this process more confidently.

### 2.3. TOM20 Proxisome Characterization as a Strategy to Identify Effectors of the Mitochondrial Co-Translational Import 

In order to enlarge the repertoire of potential co-translational import actors, we designed a strategy to identify, without any a priori, the molecular actors of this process. With the reasoning that such actors must be found in the close vicinity of the main entry gate for mitochondrial proteins (i.e., the TOM complex), the next step was thus to construct specific TOM-related BioID cell lines. To achieve this goal, we modified the genome of HCT116 cells to fuse the C-terminal part of the TOM20 receptor to a highly efficient modified biotin ligase, called mini-turboID (mTb). While performing a proximity labeling assay such as BioID, the importance of an appropriate control is crucial in order to be able to discriminate true candidates from background noise. Therefore, and as previously described [1], we also constructed a TOM20-T2A-mTb control cell line, taking the advantage of the ribosomal skipping activity of the T2A linker [54] to obtain a cytosolic mTb that is expressed at the same level as the endogenous TOM20 protein. Additionally, we constructed a third BioID cell line to obtain a more stringent control by fusing the mTb biotin ligase sequence to the gene encoding the outer mitochondrial membrane protein carnitine palmitoyl transferase 1A (CPT1A) (Appendix A). This protein was selected because of its lack of reported interaction with the TOM complex, according to the STRING database. Using this mitochondrial BioID control cell line should allow us to discriminate the proteins that are specifically found close to the TOM complex from the proteins found in proximity of the mitochondria. Following the generation of the three BioID cell lines, the expression of all three fusion proteins was monitored by Western blotting, after different times of incubation with biotin, to ensure that the biotinylating activity did not modify the stability of the fusion protein over time (Appendix A). For each cell line, both wild type and fused forms of the proteins could be detected, as expected for CRISPR-Cas9 mediated knock-in, yielding mainly heterozygous cell lines. The use of the commercially available BirA* antibody did not allow the detection of the miniTb variant. Nonetheless, for the TOM20-T2A-mTb cell line, both cleaved and complete constructs were detected with the TOM20 antibody. For the TOM20-mTb cell line, the fusion protein seemed to be expressed at the same level as the wild type allele, validating the endogenous expression of the construct. Unfortunately, the CPT1A-mTb protein could not be detected, but the abundance of the wild type protein seemed to be lower (by about 50%), suggesting either a reduced expression of the fusion protein or a reduced capacity of the antibody to recognize the modified protein (Appendix A). Since the fusion of the mTb protein could still potentially affect the endogenous function of the fused proteins, the use of heterozygous cell lines could be an advantage to reduce this potential effect on both mitochondria and cell physiology. As the overexpressed, highly efficient miniTurbo biotin ligase was previously demonstrated to actively biotinylate the targets within 10 min of incubation [21], we initially tested several short incubation times in the presence of biotin. However, we observed that a 24 h incubation time with biotin was required to ensure sufficient biotinylation (Appendix A). This discrepancy with the original report of miniTurbo biotin ligase might be attributed to the fact that in our experimental conditions, the expression level is more comparable to the endogenous expression level of only one *TOM20* allele, while it was overexpressed in the previous study [21] (Appendix A). After an incubation of 24 h in the presence of biotin, the three different cell lines showed different intensities of biotinylation (Appendix A) and the three biotinylating patterns of the different cell lines were mitochondrial, as biotinylated proteins co-localize with the TOM20 protein (Appendix A). As a cytosolic biotinylation pattern was expected for the T2A-mTb, this result was unexpected and suggested a low ribosome skipping activity of the T2A sequence in our hands. Therefore, we decided to use the CPT1A-mTb as the main control cell line for BioID analyses.

After a 24 h biotinylation time, biotinylated proteins were pulled down, digested with trypsin and analyzed by mass spectrometry. A label-free enrichment analysis was then performed on the 999 identified proteins, highlighting 597 proteins that were ≥2-fold enriched in the TOM20-mTb condition when compared to the CPT1A-mTb condition (Figure 3A). A gene ontology (GO) term analysis was then performed on these enriched proteins using the DAVID bioinformatic resource [55]. Strikingly, when evaluating GO terms related to molecular function, we observed a strong and significant enrichment in the RNA-binding function (Figure 3B), in line with the fact that RNA-binding proteins are among the best candidates to mediate mitochondrial co-translational import function [32]. Importantly, a further analysis for GO terms related to biological processes revealed that the translation GO term was significantly enriched in the BioID dataset (Appendix A) wherein several ribosomal proteins and translation-related proteins were ≥2-fold enriched (Appendix A). As a positive control, core members of the TOM complex were also among the most significantly enriched proteins (Appendix A). The AKAP1 and LARP4 proteins were also significantly enriched in the TOM20 proxisome (Figure 3A), further supporting a putative function in mitochondrial co-translational import.

### 2.4. TOM20-mTb-Based BioID Unravels New Proteins Localized in the Mitochondria

Intriguingly, among the top significantly enriched proteins in the TOM20 proxisome, we found three RNA-binding proteins reported to be exclusively nuclear: the Mediator complex subunit 15 (MED15), the Cleavage and Polyadenylation Specific Factor 2 (CPSF2) and the G-Patch Domain Containing 4 (GPATCH4) (Figure 3A). CPSF2 and GPATCH4 were even only detected in the TOM20-mTb condition and were given an arbitrary value of 64-fold change (this value corresponds thus to apparition case in the TOM20-mTb condition with no detection in the CPT1A-mTb condition), whereas MED15 was 10 times enriched in the TOM20-mTb condition when compared to the CPT1A-mTb condition. In order to consolidate these results, the subcellular localization of the three proteins was assessed by immunofluorescence and super-resolution confocal microscopy analysis, as shown in Figure 4. Transect analyses of MED15, CPSF2 and GPATCH4 (Figure 4A,B) showed a clear mitochondrial subcellular localization in NHDFs, in addition to their nuclear localization. Colocalization analyses of the three proteins with the matrix mitochondrial marker mtHSP70 were performed, using AKAP1-mtHSP70 as a mitochondria-located control and LARP4-mtHSP70 as a mitochondrially-enriched cytosolic protein (Figure 4C). Due to the combination of the background noise in the fluorescence signal for AKAP1 and the use of stringent thresholds for fluorescence intensity measurements, arbitrarily set up for the colocalization analysis, the colocalization proportion of AKAP1 with the mtHSP70 signal is only 36%. In addition, while the colocalization of the three proteins with mtHSP70 was not as strong as for the AKAP1 positive control, it was higher than the colocalization of LARP4 with mtHSP70, and significantly different for MED15 and GPATCH4 (Figure 4C). A similar mitochondrial localization of the three proteins was observed in HEK293T cells (Appendix A) and in HCT116 cells (Appendix A). In conclusion, these colocalization analyses support the mitochondrial localization of the three nuclear proteins suggested by the TOM20-BioID. To further determine the submitochondrial localization of the three proteins, super-resolution microscopy analyses would be required.

### 2.5. TOM20-mTb as a Tool to Detect Protein Entry Inside Mitochondria

The identification of nuclear proteins, among the most-enriched proteins of the TOM20 proxisome, was surprising at first and prompted us to conduct a deeper analysis of the dataset. The GO term enrichment analysis performed on all the proteins enriched ≥ 2-fold in the TOM20-mTb condition showed a highly significant enrichment of several mitochondria-related cell components with “mitochondrion”, “mitochondrial matrix” and “mitochondrial inner membrane” as the most significantly enriched GO terms (Figure 5A). Indeed, while looking at the subcellular localization of the ≥2-fold enriched mitochondrial proteins, 36.7% of proteins are mitochondrially annotated with half of those being found in the matrix of the mitochondria and almost 30% coming from the inner mitochondrial membrane, according to the MITOCARTA database [56] (Figure 5B). For the rest, the second main subcellular localization with 26.8% is, as expected, the cytoplasm, and, interestingly, 16.6% of the enriched proteins are annotated as nuclear proteins. Those proteins may not be mere contaminants or aspecifically detected proteins but could also have a mitochondrial subcellar localization, in agreement with our results for MED15, CSPF2 and GPATCH4. This high enrichment of mitochondrial proteins in our BioID data is not surprising considering the function of TOM20 as the main receptor of the TOM complex, the main entry gate for mitochondrial proteins [26]. Therefore, the BioID data presented here bring substantial and additional information to the mitochondrial proteome of HCT116 cells. While several groups have previously characterized the mitochondrial proteome using APEX-based BioID in HEK293T cells [5,14], this work conducted in HCT116 is the first evidence of endogenous TOM20-BioID screening, providing data of the mitochondrial proteome in human cells. The endogenous feature of this BioID may explain the relatively low number of mitochondrial proteins identified (~20% of the mitochondrial proteome) compared to the ~60% typically identified using BioID proximity labeling based on overexpression approaches [5,14]. However, the proportions of the different submitochondrial proteomes identified are perfectly in line with the proportions of the sub-mitochondrial proteomes, according to the MITOCARTA database (Figure 5C). Gene set enrichment analysis (GSEA) further confirmed the relationship between the enrichment of mitochondria-related GO terms and the enrichment in the TOM20-mTb condition, as shown with the emapplot performed on the whole BioID data (both negatively and positively enriched proteins) (Figure 5D). Similarly, the density plot focusing on cell components only reveals a positive enrichment of mitochondria-related GO terms (Figure 5E). Interestingly, for almost all these mitochondria-related GO terms, the most significant enrichment of proteins corresponds to proteins only detected in the TOM20-mTb condition, which were given an arbitrary value of a 64-fold change (log_2_(6)). Indeed, while looking into the subcellular distribution of the TOM20-mTb-specific proteins, almost 70% of those are mitochondrial proteins, mainly proteins in the matrix and inner mitochondrial membrane proteins (Appendix A). This underlines a strong positive correlation between the enrichment value of the different proteins and their mitochondrial localization. There may thus be a correlation between the enrichment of the protein in the TOM20-mTb condition and its entry rate inside the mitochondria, as already mentioned above. The entry rate of mitochondrial proteins is based on the balance between protein degradation and synthesis, which are also the main components directing and defining a protein’s half-life [57]. We thus postulated a correlation between the enrichment value of the mitochondrial proteins in our BioID dataset and their half-life. Taking advantage of the published half-life data of HCT116 proteins [58], we looked for a potential correlation with our own enrichment data (Appendix A). However, due to the limited half-life data available for mitochondrial HCT116 proteins (only five values matching our data), the additional half-life data of mitochondrial proteins generated from human primary hepatocytes [59] were added to strengthen the correlation (Appendix A). A significant negative correlation was reached (R = −0.24, *p* = 0.039), supporting a correlation between the enrichment of proteins in the TOM20-mTb condition, corresponding roughly to mitochondrial protein entry rate and the half-life of those proteins. The existence of such a correlation potentially extends the use of the endogenous TOM20-mTb construct to predict the half-life of mitochondrial proteins in different cell lines, in addition to the identification of new mitochondrial proteins.

## 3. Discussion

Despite evidence for mitochondrial co-translational import, the process remains poorly characterized in human cells, whereas the process was already described in different organisms [32]. Indeed, the initial publication supporting the co-translational import of OTC, ARG2 and ALDH2 into HeLa mitochondria was the first piece of evidence supporting its existence in humans, but no effector was identified [45]. In this initial study, the authors used DHFR-based reporters stabilized or not with methotrexate and described the co-translational import of the three reporters using Western blot and immunofluorescence analyses. By taking advantage of the MTS of those proteins and the specific stabilization capacity of trimethoprim toward *E. coli* DHFR [52], we constructed similar tools to assess both mitochondrial post- and co-translational imports. However, while both preARG2-DHFR and preALDH2-DHFR were described to be imported by a co-translational mechanism, we observed a post-translational import of those constructs in our experiments [45]. Several hypotheses can be proposed to explain this discrepancy. First, no information about the exact sequences that were used to generate the original reporters was available, and we cannot exclude that the additional first ten amino acids of the mature protein that were added in this study, downstream of the pre-sequence, might have an impact. Second, micrographs in the original paper were devoid of mitochondrial labeling, making colocalization of the construct with the mitochondrial network difficult to appreciate. Moreover, they used Western blot analysis to support a mitochondrial localization of the plasmid based on the slight shift of the band corresponding to the removal of the MTS associated with the construct, which is difficult to observe since the molecular weight of the MTS peptide is no more than ~3 kDa [45]. Nonetheless, we confirmed, in three different cell lines, the mitochondrial co-translational import of the preOTC-DHFR construct and demonstrated a dual post- and co-translational import for the preCOX4I1-DHFR construct. Interestingly, the regulation of mitochondrial protein import participates in the assembly of human complex IV [49,50,60]. Most importantly, translational regulation is known to be essential to ensure the coordinated assembly of mitochondrial complexes of dual genetic origin, such as complex IV [61]. Therefore, a co-translational import of nuclear-encoded complex IV subunits is expected as the import is tightly linked to translation in this case, as we observed, at least partially, for COX4I1. It would be interesting to determine whether other nuclear-encoded members of complex IV are also entirely or partially co-translationally imported or not. 

The use of specific co-translational import reporters is however limited to validate potential effectors of the mitochondrial co-translational import. We were unable to demonstrate a mitochondrial co-translational import function, using the preOTC-DHFR reporter, for AKAP1 and LARP4, although these two proteins were indicated by the literature as strong candidates for this function. Indeed, both proteins have already been shown to promote the localized translation of mitochondrial proteins at the surface of mitochondria [53]. In addition, the *Drosophila* orthologs of AKAP1 and LARP4 were also described to mediate localized translation of nucleus-encoded mitochondrial transcripts at the surface of mitochondria and the authors proposed that both proteins might favor co-translational import [42]. This proposed function in human cells is still reinforced by our result demonstrating that both proteins interact and are enriched in the proximity of TOM20. As some flexibility between co-translational and post-translational import pathways seems to exist, suggested by the mixed behavior of the preCOX4I1-DHFR reporter, we suspected that our co-translationally imported construct could have been imported post-translationally in cells LARP4 KO. However, even though the post-translational import pathway was intact, the trimethoprim did not impact the mitochondrial localization of the preOTC-DHFR reporter. A hypothesis would be that the contribution of AKAP1-LARP4 to mitochondrial co-translational import is restricted to specific mitochondrial proteins, such as SDHA, for which translation is disrupted upon AKAP1 KO [53], while OTC translation is not affected. It would thus be interesting to construct a preSDHA-DHFR reporter and to check if it is co-translationally imported using trimethoprim, in LARP4 wild type (WT) and knock-out (KO) cells. Another possibility could be that compensatory mechanisms may exist and that co-translational effectors may be redundant. Altogether, the use of only one specific co-translational reporter is probably insufficient to assess the mitochondrial co-translational import function of identified candidates, and additional ones should be constructed and characterized. Importantly, the different reporters used in our study are all based on the MTS of the mitochondrial proteins, which can be a limitation since the MTS is probably not the only sequence triggering and promoting the co-translational import of the corresponding protein. Indeed, other sequences of the proteins or untranslated regions of the mRNA may be important/essential for the co-translational import process. As an example, the yeast RNA-binding protein Puf3 binds specific nucleus-encoded mitochondrial transcripts through their 3′UTR and ensures their localized translation at the surface of mitochondria, favoring their co-translational import [33]. Therefore, constructs based on the fusion of the full-length proteins as well as constructs containing the untranslated regions of the corresponding transcripts could result in a different and more relevant outcome and should be tested.

The use of a modified biotin ligase at the surface of mitochondria has already been described for the identification and characterization of proteins and transcripts located at the surface of the yeast and human mitochondria [7,14,16,17] but not for the identification of proteins that are specifically in the close (within a 10 nm range) vicinity of the TOM complex, the main entry gate for mitochondrial proteins [26]. TOM20 protein was an interesting candidate to generate a fusion protein with the biotin ligase because of its key role in the pre-sequence pathway for mitochondrial protein import, and because of the fact that the yeast TOM20 (Tom20p) has previously been shown to facilitate the co-translational import of mitochondrial proteins [26,39]. Moreover, the endogenous tagging in the C-terminal part of TOM20 protein does not change the mitochondria physiology, as a TOM20-EGFP fusion does not impair the potency of iPSCs, with the mitochondria being essential for stem cell differentiation [62,63]. This is further supported by the size of the miniTurbo enzyme, which is even smaller than the GFP [21]. However, the endogenous expression of the miniTurbo enzyme also demonstrated that a longer and more typically used biotinylation time of 24 h was required when compared to overexpression conditions in which 10 min of biotinylation is enough to obtain a strong biotinylation of surrounding proteins [21]. Interestingly, more recent variants of the BirA* protein, microID and ultraID, were developed recently and show an even higher labeling efficiency, with a 10 min biotinylation time shown to be sufficient, even under physiological expression [22].

The use of carefully selected controls is of uttermost importance in proximity-labeling assays such as BioID experiments. Indeed, due to endogenous biotin-binding proteins and to unspecific protein binding associated with pull down experiments, proper controls are required to discard false positives naturally present in proximity-labeling assays. Moreover, in BioID experiments, the labeling range is believed to be within 10 nm, and therefore it is essential to use stringent controls to filter and narrow the scope down to the proteins found really close to the bait protein, within an acceptable range of interaction [11,64]. While working endogenously, the use of a self-cleavable T2A linker was particularly relevant as it allowed the use of the same fusion protein—and the same expression level of this protein—for the control cell line compared to the experimental cell line. This experimental design was first reported as an elegant proof of concept for the identification of endogenous p53 proxisome in HCT116 cells [1]. Unexpectedly, the ribosome-skipping activity of the T2A peptide seemed to be low in our TOM20-T2A-mTb cell line, as suggested by the absence of cytosolic biotinylation and the mitochondrial biotinylating pattern, which was very similar to the one observed in the TOM20-mTb cell line. This result was surprising regarding the high cleavage efficiency (almost 90%) of the T2A linker reported historically [54]. However, a reduced protein expression downstream of the T2A sequence has been reported using T2A-GFP constructs in HEK293T and might explain the really low cytosolic biotinylating activity of this construct [65]. Unfortunately, this hypothesis could not be confirmed due to the low sensitivity of the BirA* antibody toward the highly modified mTb variant and would require more specific antibodies. Therefore, we decided to work with the additional CPT1A-mTb control, even though the expression of the construct seems to be lower when compared to the TOM20-mTb. The CPT1A-mTb control has the advantage of being mitochondrial but without the described interaction with the TOM complex, according to the STRING database.

The identification of the TOM complex proxisome using endogenous TOM20-mTb revealed a strong enrichment in RNA-binding proteins, in line with the enrichment of nucleus-encoded mitochondrial transcripts at the surface of mitochondria, already described in human cells [16,46,47]. Moreover, RNA-binding function is already known to be important for the co-translational import and thus indicates that the TOM20 proxisome could be an important resource for the identification of mitochondrial co-translational import effectors. Moreover, the high enrichment of ribosome proteins and translation initiation factors supports the presence of actively translating ribosomes close to the TOM complex. Interestingly, a direct interaction between ribosomes and the TOM complex has been characterized in yeast using electron cryo-tomography [66]. In addition, the interaction of AKAP1 and LARP4 is already known to promote localized translation at the surface of mitochondria [53] and we showed here their close vicinity to the TOM complex. While Vardi-Oknin and Arava detected CLUH using proximity-specific ribosome profiling, we were unable to detect the protein in our BioID analyses. This may be explained by the cell type used in our present study (HCT116), different from the one used in Vardi-Oknin and Arava’s study (HEK293T) [18]. Alternatively, more recent results seem to indicate that CLUH would be mainly located in stress granule-like structures where it would exert its translational regulator function [67]. Therefore, even though this protein regulates the translation of nucleus-encoded mitochondrial transcripts, it would not be involved in the regulation of their co-translational import. This is in line with the results obtained here and the apparent absence of CLUH at the surface of the mitochondria. In conclusion, the BioID dataset provided in this study could contribute to identify effectors of the mitochondrial co-translation import in human cells, but this would require first the characterization and the use of additional specific co-translational reporters.

Besides the strong potential of TOM20-based BioID in the assessment of human mitochondrial co-translational import, the localization of miniTurbo in the close vicinity of the main entry gate for mitochondrial proteins is also relevant for the identification of proteins entering mitochondria. Indeed, the high enrichment of mitochondrial proteins in the TOM20-mTb compared to the control suggests that a vast majority of the enriched proteins correspond to proteins entering mitochondria, corroborated by the strong mitochondrial biotinylating signal observed in TOM20-mTb cells. In addition, the GSEA analysis and the analysis of protein localization showed that the stronger enrichment in the TOM20-mTb condition implies an increased likelihood of identifying a bona fide mitochondrial protein, and, more specifically, a protein of the mitochondrial matrix. Our BioID dataset could thus also be used as a representative mitochondrial proteome of HCT116 while considering all the ≥2-fold enriched mitochondrial proteins. Indeed, even though only 20% of the whole mitochondrial proteome was identified in this work, the distribution of the mitochondrial proteins among the different sub-compartments parallels the endogenous repartition documented in the MITOCARTA database. In line with this observation, we could identify proteins reported to be exclusively nuclear inside the mitochondria in at least three different cell lines. Indeed, MED15, a transcription-related protein involved in cholesterol metabolism [68,69,70], CPSF2, involved in polyadenylation of mRNAs [71,72] and GPATCH4, involved in rRNA maturation [73], were never demonstrated to be localized outside the nucleus. Interestingly, another mediator subunit, MED12, was already detected inside the mitochondria in HEK293T cells [74]. The colocalization of the three proteins with the matrix marker mtHSP70 was higher compared to LARP4, known to be enriched at the mitochondrial surface, confirming the correlation between high enrichment and potential mitochondrial localization. Additional nuclear proteins, such as NUP62 or ADIRF, were detected among the highly enriched proteins and it may be of interest to confirm their subcellular localization too. The interest in generating a representative mitochondrial proteome based on this approach, rather than classical approaches such as mass spectrometry on mitochondrial fractions or APEX-based proximity labeling, is due to the addition of a time dimension with a biotinylating time window that is much longer. In the 24 h time frame of the TOM20-BioID, no sub-compartment is overrepresented, supporting a homogenous entry of mitochondrial proteins over time in terms of sub-localization. The entry rate of mitochondrial proteins is reminiscent of the balance between protein degradation and synthesis and therefore is correlated with the protein’s half-life [57]. Interestingly, we could obtain a significant negative correlation between the enrichment value for the matrix mitochondrial proteins, directly dependent on TOM20-mediated import, and their reported half-life. However, facing the paucity of half-life data for mitochondrial proteins in the HCT116 cell line [58], we took data generated for primary hepatocytes [59], but a high variability does exist between half-life values for different cell types [59,74]. Therefore, before using the TOM20-mTb cell line as a predictive tool for mitochondrial protein half-life, we should first evaluate the HCT116 half-life data of mitochondrial proteins using classical pulse-chase labeling and mass spectrometry [57,59] in order to verify the correlation. In addition, it is also of importance to validate that the direct biotinylation of mitochondrial proteins has no effect on their half-life time. Indeed, we cannot exclude that the addition of biotin moieties on proteins might impair their ability to assemble into multimolecular complexes, and it has been reported that unassembled subunits of larger complexes are quickly degraded, as demonstrated for mitoribosomes and respiratory chain complexes [75,76]. Therefore, a first step before validating the application of the TOM20-mTb cell line as a predictive tool for mitochondrial protein half-life time would be to validate that the biotin moiety does not affect the potential of proteins to assemble into larger complexes. This could be achieved, as previously described [75,76], by using pulse-chase SILAC-based analyses of mitochondrial ribosomal proteins or complex I subunits, in the TOM20-mTb cell line.

In conclusion, several applications of a TOM20-mTb cell line can be proposed for the study of human mitochondrial proteins. We first showed, as a proof of principle, an experimental design aiming at the unbiased identification and validation of mitochondrial co-translational import effectors. Moreover, the TOM20-based BioID showed an additional and/or alternative utility for the tracking of proteins entering mitochondria. This could be used for the identification of new mitochondrial proteins or for mitochondrial protein half-life prediction.

## 4. Materials and Methods

### 4.1. Isolation of Primary Fibroblasts

Primary normal human dermal fibroblasts (NHDFs) were isolated from young foreskin samples as previously described [77]. Samples were obtained after circumcision (Dr L. de Visscher, Clinique St-Luc, Bouge, Belgium) following approval by the Ethics Committee of the Clinique St-Luc (Bouge, Belgium).

### 4.2. Cell Culture

The HeLa, HCT116 and HEK293T cell lines were maintained in Dulbecco’s Modified Eagle Medium (DMEM) with a high glucose concentration of 4.5 g/L and supplemented with 10% fetal bovine serum (FBS) (ThermoFisher Scientific, Waltham, USA), 1% penicillin–streptomycin (ThermoFisher Scientific) at 37 °C and 5% CO_2_. NHDFs were maintained in Basal Medium Eagle (BME) (ThermoFisher Scientific) supplemented with 10% FBS, 2 mM L-Glutamine (ThermoFisher Scientific) and 1% penicillin–streptomycin. Expanding cells were maintained under 80% confluency and passed by using trypsin-EDTA (ThermoFisher Scientific).

### 4.3. Generation of KO Cell Lines

KO of LARP4 in HEK293T cells was performed using CRISPR-Cas9 technology. Briefly, crRNA (5′-AATTTGGACAGTTGCCAACA-3′) (Integrated DNA Technologies (IDT), Coralville, USA) targeting exon 5 of the *LARP4* gene was designed using CRISPOR bioinformatic resource [78]. An amount of 150 picomoles of crRNA was annealed 1:1 with a universal tracrRNA (IDT) by cooling it down from 95 °C to 30 °C at a rate of 5 °C per minute. The crRNA–tracrRNA heteroduplex was then mixed with 10 µg of recombinant Cas9 (IDT) for 10 min at room temperature. Meanwhile, 1 million cells were harvested and resuspended in nucleofector solution from Nucleofector Kit V (Lonza, Bale, Suisse), supplemented with the ribonucleoparticle of crRNA–tracrRNA and Cas9. Cells were next electroporated using Amaxa Nucleofector Ib with the C-09 program and plated in 6-well plates with prewarmed medium. After 72 h, a limiting dilution was performed and single cells were transferred in a 96-well plate. The clones were then expanded and two WT (clones 2 and 3) and two KO (clones 9 and 10) clones were selected based on Western blot analysis and LARP4 abundance assessment.

### 4.4. Generation of Endogenous BioID Cell Lines

TOM20-T2A-miniTurbo, CPT1A-miniTurbo and TOM20-miniTurbo HCT116 cell lines were obtained using CRISPR Cas9-mediated knock-in (KI). crRNAs (IDT) targeting the genomic sequence just before the stop codon of the *TOMM20* and *CPT1A* genes were designed using CRISPOR. An equivalent of 1 million HCT116 cells were harvested and resuspended in nucleofector solution from Nucleofector Kit V (Lonza), supplemented with the preassembled ribonucleoparticle of crRNA–tracrRNA and Cas9 and with 2 µg of either TOMM20Cter-T2A-miniTurbo, CPT1Acter-miniTurbo or TOMM20Cter-miniTurbo template repair plasmids (see below). Cells were electroporated using Amaxa Nucleofector Ib with the C-09 program and plated in 6-well plates with prewarmed culture medium. After 72 h, a limiting dilution was performed and single cells were transferred into a 96-well plate. Clones were expanded and one heterozygous HCT116*^WT/BioID^* clone for each cell line was selected following a PCR screening in the 96-well plate, as previously described [79].

### 4.5. Repair Template Plasmids and Reporters Cloning

The template repair plasmids used for the generation of the endogenous BioID cell lines were constructed based on the pAav_TP53cter-T2Aopt-miniTurbo, for the TOM20-T2A-miniTurbo cell line, and pAav_TP53cter-miniTurbo, for the CPT1A-miniTurbo and TOM20-miniTurbo cell lines, kindly provided by Pr. Sven Eyckerman. The TP53 homology regions (HR) of the plasmids were replaced by homology arms of 1 kb upstream (5′HR) and downstream (3′HR) of the cleavage site at the end of the last coding exon of *TOMM20* (Refseq: ENSG00000173726) and *CPT1A* (Refseq: ENSG00000110090). The 5′HR cloning was ordered and made by Genscript (Piscataway, USA). The 3′HR for the CPT1A-based construct was picked by PCR from genomic DNA of HCT116 while the 3′HR of TOM20-based constructs was picked by PCR from the donor plasmid as described in [63] (Addgene plasmid # 87423; http://n2t.net/addgene:87423 accessed on 6 October 2022; RRID:Addgene_87423).

All used reporters (preALDH2-DHFR, preARG2-DHFR, preCOX4I1-DHFR and preOTC-DHFR) were ordered and made by Genscript, based on the ecDHFR-myc reporter (Addgene plasmid # 20214; http://n2t.net/addgene:20214 accessed on 6 October 2022; RRID:Addgene_20214) [80] to which the mitochondrial targeting sequence of each protein with the 10 following amino acids was added in frame at the N-terminal part of the ecDHFR coding DNA sequence, depleted for its start codon (Appendix A).

### 4.6. Proximity Labeling

BioID cell lines were incubated for 24 h with medium supplemented with 50 µM biotin. The medium was then changed to remove biotin for 3 h before cell lysis. Cells were harvested, resuspended in RIPA buffer and incubated for 45 min on a thermomixer (Eppendorf) at 4 °C and 1000 RPM (Round Per Minute). The lysates were then sonicated three times for 10 s at 70% amplitude, frequency 1, and centrifuged for 15 min at 16.000× *g* at 4 °C to collect supernatants. Sample protein content was determined with the Pierce detection assay and a maximum amount of proteins (2.3 mg) was loaded on 1 mg of Dynabeads M-280 Streptavidin (ThermoFisher Scientific), previously rinsed with RIPA buffer. An equivalent of 2.5% of total proteins (57.5 µg) was saved as the input for Western blot analysis. Protein lysates were incubated with the beads on a rotating wheel at 4 °C for 16 h. The beads were then washed according to Le Sage and colleagues [81]. Briefly, proteins were successively washed with washing buffer 1 (2% SDS in sterile deionized water), washing buffer 2 (50 mM HEPES; pH 7.5, 500 mM NaCl, 1 mM EDTA, 0.1% sodium deoxycholate, 1% Triton X-100) and washing buffer 3 (10 mM Tris-HCl pH 7.4, 250 mM LiCl, 1 mM EDTA, 0.1% sodium deoxycholate, 1% NP-40). Finally, the beads were washed twice with 20 mM Tris-HCl (pH 8) and resuspended in 20 µL of 20 mM Tris-HCl (pH 8). About 5% of the pull-down volume was saved for Western blot analyses. Beads were then digested with 1 µg of trypsin gold (Promega) overnight at room temperature in a thermomixer at 600 RPM. Supernatant was then transferred to a new vial and 500 ng of trypsin was added followed by a 3 h incubation at room temperature in a thermomixer at 400 RPM. Trypsin was then inactivated with 2% formic acid (Biosolve). The digested peptides were then rinsed using Pierce C18 Spin Tips & Columns system according to the manufacturer’s instructions (ThermoFisher Scientific). Three independent biological replicates/samples for each condition were analyzed by mass spectrometry.

### 4.7. Mass Spectrometry Analyses

Digested proteins were analyzed using nanoLC-ESI-MS/MS tims TOF Pro (Bruker, Billerica, USA) coupled with an UHPLC nanoElute (Bruker). Peptides were analyzed by nanoUHPLC (nanoElute, Bruker) on a 75 μm ID, 25 cm C18 column with integrated CaptiveSpray insert (IonOpticks, Victoria, Australia) at a flow rate of 200 nL/min, at 50 °C. LC mobile phase A was 0.1% formic acid (*v/v*) in water and B was formic acid 0.1% (*v/v*) in acetonitrile. Peptides (1 µL) were injected, and the gradient was increased from 2% B to 15% in 40 min, from 15% B to 25% in 15 min, from 25% B to 37% in 10 min and from 37% B to 95% in 5 min. Data acquisition on the timsTOF Pro was carried out using Hystar 5.1 and timsControl 2.0. Data were acquired using 160 ms TIMS accumulation time, mobility (1/K0) range from 0.75 to 1.42 Vs/cm^2^. Mass spectrometric analysis was performed using the parallel accumulation serial fragmentation (PASEF) acquisition method [82]. Cycles of one MS spectra followed by six PASEF MSMS spectra in a total duration of 1.16 s were carried out. Two injections per sample were performed.

Data analysis was carried out using PEAKS Studio 11 X Pro with ion mobility module and Q module for label-free quantification (Bioinformatics Solutions Inc., Waterloo, Canada). PEAKS software was used to identify the proteins with 15 ppm as parent mass error tolerance and 0.05 Da as fragment mass error tolerance. The oxidation of methionine, biotinylation of lysine and acetylation (N-term) were allowed as variable modifications. Trypsin was used as a digestion enzyme and the maximum number of missed cleavages per peptide was set at two. The protein library used was Homo Sapiens with isoforms from UNIREF 100 (195195 sequences) and the sequence of the miniTurbo protein was added. Peptide spectrum matches and protein identifications were normalized to less than 1.0% false discovery rate. 

The label-free quantitation (LFQ) method is based on the expectation–maximization algorithm on the extracted ion chromatograms of the three most abundant unique peptides of a protein to calculate the area under the curve [83]. Mass error and ion mobility tolerance were set, respectively, to 15 ppm and 0.08 1/k0 for the quantitation. Peptide quality score was set to be ≥ 3 and the protein significance score threshold was set to 15. The significance score is calculated as the −10log_10_ of the significance testing *p*-value (0.05), with the ANOVA used as the significance testing method. Total ion current was used for the normalization of each extracted ion current. Only two replicates of the TOM20-mTb condition could be used for quantification analysis because of the poor quality of the third replicate showing high deviation compared to the two others.

The exported PEAKS data of label-free quantification were sorted and represented using R v4.3.0 software with the “EnhancedVolcano” package [84]. The fold change cutoff was set at 2 and the significance cutoff was set at 1 × 10^−15^. 

The mass spectrometry proteomics data have been deposited to the ProteomeXchange Consortium via the PRIDE [85] partner repository, with the dataset identifier PXD038821 and 10.6019/PXD03882. Data can be accessed using the username reviewer_pxd038821@ebi.ac.uk and the password p59LtaI8.

### 4.8. Cell Transfection with Reporter Constructs

HCT116 and HEK293T cells were seeded on coverslips (n = 1.5, VWR) in 24-well plates at 45,000 cells/cm^2^. After 24 h, the cells were transfected with 150 ng of reporters (Genscript), preALDH2-DHFR, preARG2-DHFR, preCOX4I1-DHFR and preOTC-DHFR, following a preincubation of 30 min with the XTremeGENE HP Transfection Reagent (Roche) in Opti-MEM I serum-free medium (ThermoFisher Scientific). After 4 h, the cell culture medium was replaced by DMEM containing 10% FBS, 1% penicillin–streptomycin medium, supplemented or not with 100 μM of trimethoprim (Sigma-Aldrich). After 24 h, cells were prepared for immunofluorescence analysis and confocal microscopy observation.

### 4.9. Immunofluorescence Analysis and Confocal Microscopy Observation

Cells were washed three times with phosphate-buffered saline (PBS), pH 7.4, 37 °C, fixed with 4% paraformaldehyde and rinsed again three times with PBS. Fixed cells were then permeabilized for 5 min with PBS-1% Triton X-100. To limit the unspecific signal, cells were rinsed twice for 10 min with PBS-2% Bovine Serum Albumin (BSA). Primary antibody solutions for immunostaining were prepared in PBS-2% BSA and incubated with fixed cells at 4 °C overnight. The primary antibodies used were rabbit polyclonal TOM20 (ab186734, Abcam, Cambridge, UK); mouse monoclonal Myc-tag (2276S, Cell Signaling Technology (CST), Danvers, USA); mouse monoclonal mtHSP70 (ALX-804-077-R100, Enzo, New-York, USA); rabbit monoclonal AKAP1 (5203S, CST); rabbit polyclonal LARP4 (ab241489, Abcam); rabbit polyclonal MED15 (HPA003179-100UL, Sigma-Aldrich); rabbit polyclonal CPSF2 (ab229114, Abcam); rabbit polyclonal GPATCH4 (ab246961, Abcam); and the streptavidin-Alexa488 probe (S32354, ThermoFisher Scientific).

The next day, the cells were rinsed twice for 10 min with PBS-2% BSA. Cells were then incubated with secondary antibody in PBS-2% BSA supplemented with 1 µg/mL of 4′,6-diamidino-2-phenylindole (DAPI) (Sigma-Aldrich) intercalating agent for 1 h at room temperature in the dark. Secondary antibodies used were goat polyclonal anti-Rabbit (Alexa Fluor 488 nm) (A-11008, ThermoFisher Scientific); goat polyclonal anti-Rabbit (Alexa Fluor 568 nm) (A-11011, ThermoFisher Scientific); and goat polyclonal anti-Mouse (Alexa Fluor 568 nm) (A-11031, ThermoFisher Scientific). Cells were then rinsed twice with PBS-2% BSA for 10 min and then left in PBS. The mounting of the slides was performed with warmed Mowiol (Sigma-Aldrich) or Fluoromount G (Invitrogen). As indicated in figure legends, confocal micrographs were acquired with a Leica SP5 Confocal Microscope (Leica microsystem) or a Zeiss LSM900 confocal microscope (Zeiss, Oberkochen, Germany). 

### 4.10. Image Analysis

Colocalization analyses were performed on Fiji software (ImageJ2) [86] using a home-made macro and the ComDet plugin [87]. The average particle sizes and the thresholds of intensities were arbitrarily set up for each cell line and for each micrograph to fit as much as possible to the observed signal for both channels.

### 4.11. Western Blotting

Cells were scraped and lysed in homemade radioimmunoprecipitation assay (RIPA) lysis buffer (10 mM Tris-HCl; pH 8.0, 1 mM EDTA, 0.5 mM EGTA, 1% Triton X-100, 0.1% Sodium Deoxycholate, 0.1% SDS, 140 mM NaCl) complemented with SuperNuclease (25 U/µL, Bio-connect), complete Protease Inhibitor Cocktail (Roche) and homemade Phosphatase Inhibitor Cocktail (25 mM Na_3_VO_4_, 250 mM 4-nitrophenylphosphate, 250 mM di-Sodium β-glycerophosphate pentahydrate and 125 mM NaF) and maintained on ice. Mixing with a thermomixer was performed for 10 min at 4 °C and 1400 RPM. Cell lysates were then centrifuged for 10 min at 16.000× *g* at 4 °C to collect supernatants. Sample protein content was determined with the Pierce detection assay (Pierce 660 nm Protein Assay Reagent (ThermoFisher Scientific), complemented with Ionic Detergent Compatibility Reagent (IDCR) (ThermoFisher Scientific) according to the manufacturer’s recommendations.

An equivalent of 20 μg of proteins was prepared in Western blot loading buffer (30 mM Tris; pH 6.8, 1.2% SDS, 3% β-mercaptoethanol, 5% glycerol and 150 µM bromophenol blue), boiled for 5 min at 98 °C and resolved on home-made polyacrylamide gel (SDS-PAGE). The Color Protein Standard Broad Range (10–250 kDa) (New England BioLabs Inc.) was used as protein molecular weight ladder. Protein electrophoresis was performed at 150–200 V (400 mA and 100 W). Proteins were then transferred on a PolyVinyliDene Fluoride (PVDF) membrane (Immobilon-P, Merck Millipore, Burlington, USA) using liquid transfer with 20% methanol at 100V, at 4 °C for 2 h. The PVDF membrane was then incubated for 1 h with the blocking solution Intercept Blocking Buffer (TBS) (Li-cor Biosciences, Lincoln, USA) at room temperature. The primary antibody solutions were prepared in the Intercept Blocking Buffer (TBS) containing 0.1% Tween-20 (Roth) and incubated overnight with the membrane on a rocker at 4 °C. The primary antibodies used were mouse monoclonal BirA (NBP2-59939, Novus Biologicals, Abingdon, UK); mouse monoclonal CPT1A (ab128568, Abcam); rabbit polyclonal Biotin (ab53494, Abcam); rabbit polyclonal LARP4 (ab241489, Abcam); and mouse monoclonal α-tubulin (TUBA4A (7277)) (926-68070, Li-cor Biosciences). The secondary antibody solution was prepared in the Intercept Blocking Buffer (TBS) with 0.1% Tween-20 and the following: anti-rabbit goat polyclonal antibody (IR Dye 800CW) (926-32211, Li-cor Biosciences) and anti-mouse goat polyclonal antibody (IR Dye 680RD) (926-68070, Li-cor Biosciences). Membrane fluorescence was detected using the Odyssey Li-cor Scanner (Li-cor Biosciences).

### 4.12. Immunoprecipitation

HEK293T cells were scraped in RIPA buffer and the lysate was incubated for 30 min on a rotating wheel at 4 °C before a centrifugation for 10 min at 16.000× *g* at 4 °C to collect supernatants. Sample protein content was determined with the Pierce detection assay and 800 µg of proteins was loaded on Dynabeads Protein G for Immunoprecipitation (ThermoFisher Scientific); the proteins were previously incubated for 10 min on a wheel at room temperature with PBS-Tween-20 0.01% supplemented or not with 5 µg of LARP4 antibody (ab241489, Abcam). In total, 5% of total protein (40 µg) was saved for input. Protein lysates were incubated with the beads on a wheel at 4 °C for 16 h. The beads were then rinsed three times with NETN buffer (50 mM Tris-HCl; pH 8, 100 mM NaCl, 1 mM EDTA, 0.5% NP40), twice with ETN (50 mM Tris-HCl; pH 8, 100 mM NaCl, 1 mM EDTA) and once with ddH_2_O. Beads were then resuspended in Western blot loading buffer and boiled for 5 min at 98 °C before resolution by SDS-PAGE (sodium dodecyl sulfate-polyacrylamide gel electrophoresis).

### 4.13. Data Analyses

All data and statistical analyses were performed using the statistical programming language R (http://www.rproject.org/ accessed on 6 October 2022). Unpaired *t*-tests for 2-by-2 comparisons and ANOVA followed by Dunnett’s post hoc test for multiple comparisons were performed after data normality verification. A GSEA analysis was performed on the BioID whole dataset using R and the “ClusterProfiler package” [88]. The analyzed and sorted BioID data are available as a Appendix A.

## Figures and Tables

**Figure 1 ijms-24-09604-f001:**
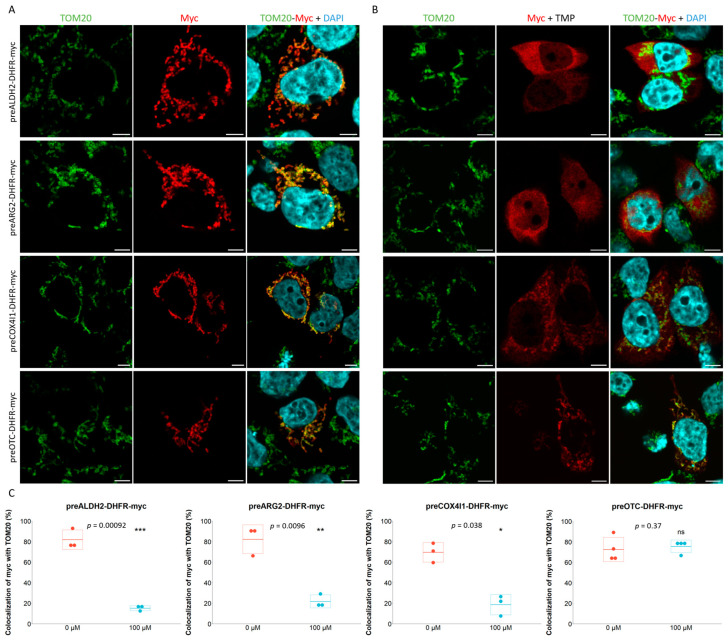
**Functional assessment of mitochondrial co-translational import reporter proteins.** (**A**). HCT116 cells were seeded and transfected 24 h after with either preALDH2-, preARG2, preCOX4I1- or preOTC-DHFR-myc plasmid. The medium was replaced 4 h after transfection with medium supplemented (**B**) or not (**A**) with 100 µM trimethoprim (TMP). Cells were fixed in 4% PFA 24 h after and labeled with Myc and TOM20 antibodies. Nuclei were stained using DAPI. Micrographs were acquired on a Leica TCS SP5 confocal microscope. Scale bar = 5 µm. (**C**). The quantifications were performed using Fiji software (ImageJ2) with the ComDet colocalization plugin. Analyses were performed on 10–20 cells in at least three biological replicates and a paired t-test was performed using R v4.3.0 software. * *p* < 0.05, ** *p* < 0.01, *** *p* < 0.001; ns: no significance.

**Figure 2 ijms-24-09604-f002:**
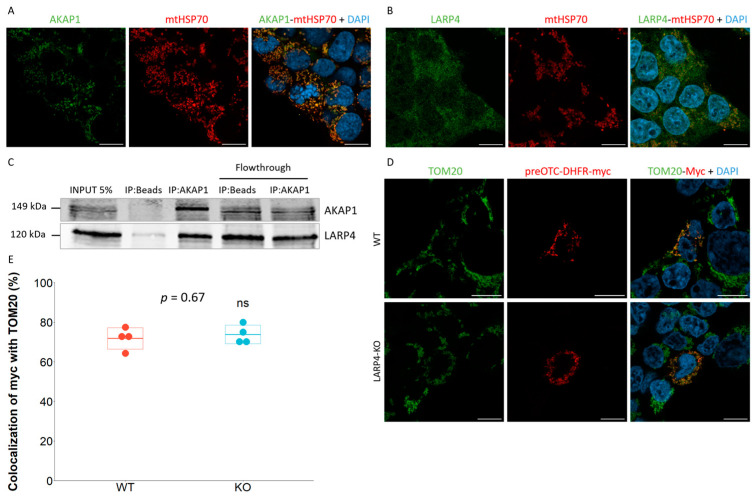
**Assessment of the AKAP1-LARP4 candidates in the mitochondrial co-translational import.** (**A**,**B**). HEK293T cells were seeded and fixed in 4% PFA 24 h after and labeled with TOM20 and AKAP1 (**A**) or LARP4 (**B**) antibodies. Nuclei were stained using DAPI. Micrographs were acquired on a Zeiss LSM900 Airyscan confocal microscope. Scale bar = 10 µm. Experiment performed on three independent biological replicates. (**C**). HCT116 cells were harvested 24 h after seeding and lysed in RIPA buffer and 1 mg of proteins was loaded on magnetic beads coupled or not with 5 µg of AKAP1 antibody. Proteins were immunoprecipitated for 16 h on a wheel at 4 °C and were then washed and resuspended in Western blot loading buffer. About 5% of the lysate (50 µg) was resolved by SDS-PAGE along with the pulled-down proteins on acrylamide gel. AKAP1 and LARP4 were revealed by Western blot analysis. (**D**). WT and LARP4 KO HEK293T cells were seeded and transfected 24 h after with preOTC-DHFR-myc reporter. The cell culture medium was replaced 4 h after transfection and cells were fixed in 4% PFA 24 h later and labeled with Myc and TOM20 antibodies. Nuclei were stained using DAPI. Micrographs were acquired with a Zeiss LSM900 Airyscan confocal microscope. Scale bar = 10 µm. (**E**). Colocalization quantifications were performed using Fiji software (ImageJ2) with the ComDet colocalization plugin. Analysis was performed on 5–10 cells in 4 independent biological replicates and a paired *t*-test was performed using R v4.3.0 software; ns: no significance.

**Figure 3 ijms-24-09604-f003:**
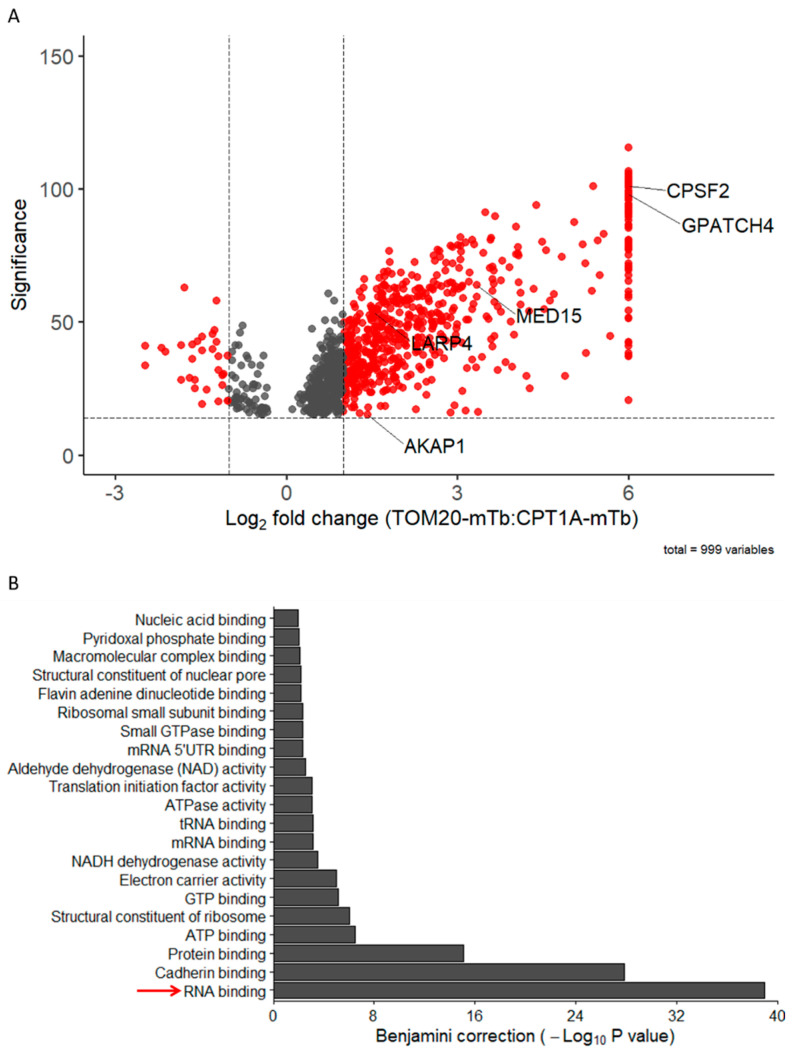
**BioID data analysis.** (**A**). Label-free quantitative analysis was performed using PEAKS Studio 11 software (Bioinformatics Solutions Inc., Waterloo, Canada) and comparing protein enrichment in the TOM20-mTb pull-down relative to CPT1A-mTb pull-down. A 1% FDR and a single peptide detection threshold were set. Peptides detected in only one of the two experimental conditions were given an arbitrary 64-fold change value (n = 2 for TOM20-mTb and n = 3 for CPT1A-mTb). Five candidates of interest are highlighted on the volcano plot. (**B**). Gene ontology enrichment analysis of the ≥2-fold-enriched proteins in the TOM20-mTb condition relative to CPT1A, using DAVID resource [55], showing the top enriched molecular function GO terms. RNA binding GO term is highlighted with the red arrow.

**Figure 4 ijms-24-09604-f004:**
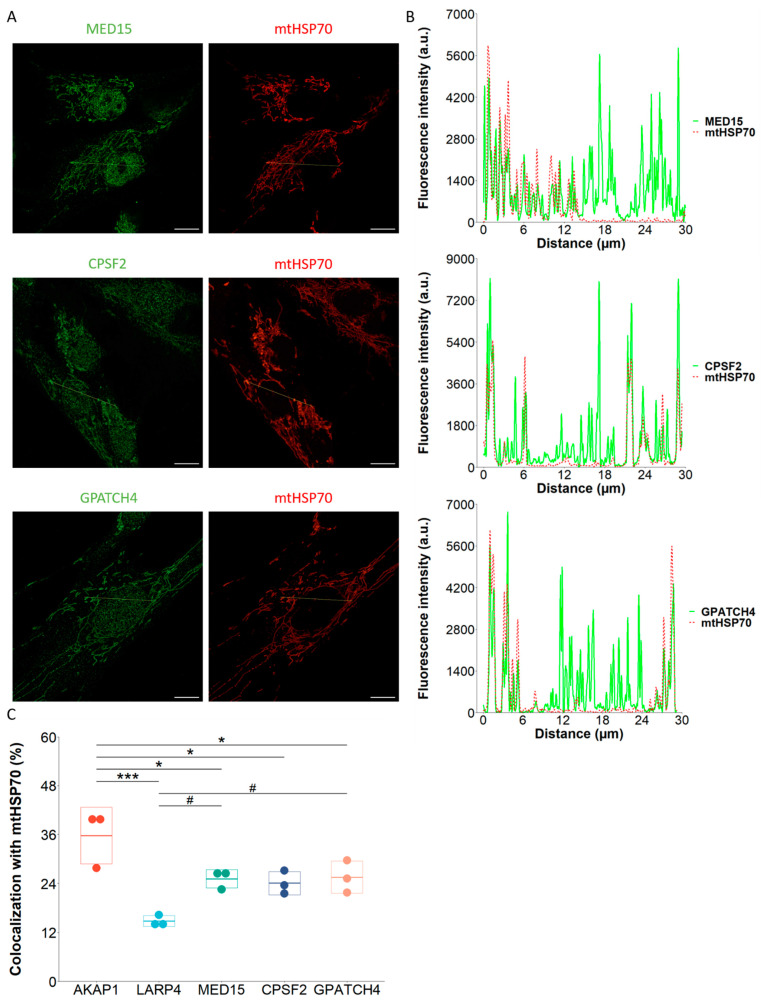
**Subcellular localization of MED15, CPSF2 and GPATCH4.** (**A**). NHDFs cells were seeded and fixed in 4% PFA 24 h after and labeled with MED15, CPSF2, GPATCH4 and mtHSP70 antibodies. Nuclei were stained using DAPI. Micrographs were acquired on a Zeiss LSM900 Airyscan confocal microscope. Scale bar = 10 µm. Experiment performed on three biological replicates. (**B**). Associated transect analyses using Fiji software (ImageJ2) and R v4.3.0 software. (**C**). Colocalization quantifications of AKAP1, LARP4, MED15, CPSF2 and GPATCH4 with mtHSP70 mitochondrial marker were performed using ComDet Fiji plugin on ~20 cells in three independent biological replicates and ANOVA followed by Dunnett’s post hoc tests was performed using R software with AKAP1 (*) or LARP4 (#) as the reference comparison group. * *p* < 0.05, *** *p* < 0.001, # *p* < 0.05.

**Figure 5 ijms-24-09604-f005:**
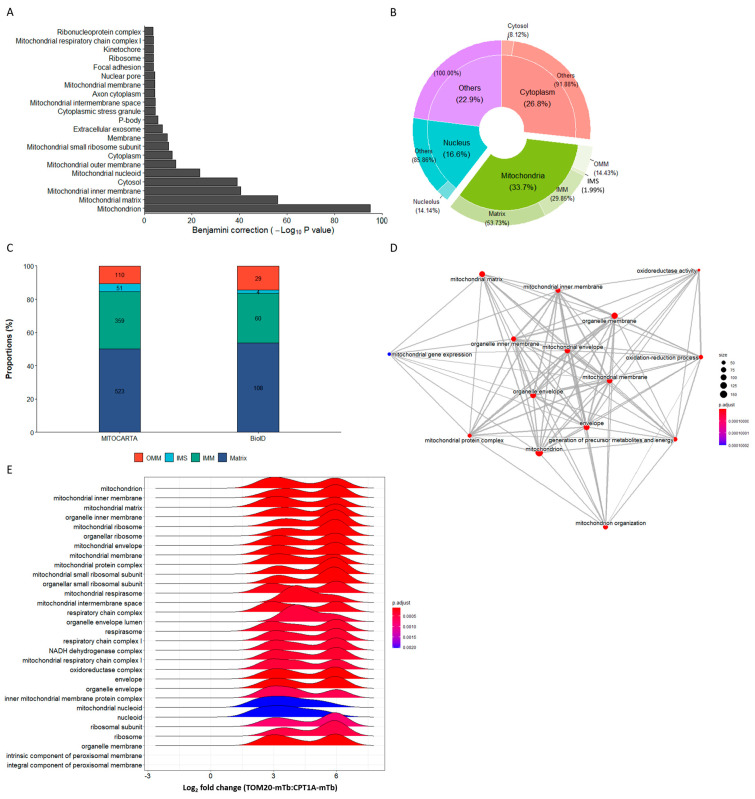
**BioID in-depth analyses reveal high enrichment of mitochondrial proteins.** (**A**)**.** Gene ontology enrichment analysis performed on the ≥ 2-fold-enriched proteins in the TOM20-mTb condition relative to CPT1A-mTb, using DAVID platform [55], showing the top enriched cell component GO terms. (**B**). Repartition of the different subcellular localizations of the ≥ 2-fold-enriched proteins in the TOM20-mTb condition relative to CPT1A-mTb. Subcellular localization data were obtained from MITOCARTA database [56] and from Uniprot (Universal Protein Resource) database. (**C**). Proportions of the different submitochondrial proteomes relative to the whole mitochondrial proteome in the MITOCARTA database and in the BioID data, after filtration of all the mitochondrial proteins ≥2-fold-enriched in the TOM20-mTb condition relative to CPT1A-mTb. (**D**,**E**). A GSEA analysis was performed on the BioID whole dataset using R and the ClusterProfiler package. (**D**)**.** The emapplot represents all the most significantly enriched GO terms of the three different families (biological process, cellular component and molecular function) and their functional links, showing a strong association of mitochondria-related GO terms. (**E**). The density plot represents the most significantly enriched cell component family of GO terms in regard to the enrichment distribution of the proteins in the BioID dataset. The enrichment value of 6 corresponds to the proteins arbitrarily attributed with a 64-fold change value based on their unique presence in the TOM20-mTb samples.

## Data Availability

All raw mass spectrometry data and export files related to Figure 2 can be found in the PRIDE database with the dataset identifier PXD038821 and 10.6019/PXD038821.

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
