# Peer review of "Endogenous TOM20 Proximity Labeling: A Swiss-Knife for the Study of Mitochondrial Proteins in Human Cells"

_ijms, 2023, doi:10.3390/ijms24119604_

Round 1

Reviewer 1 Report (Previous Reviewer 1)

The authors responded to all my comments and fulfilled all requests. The manuscript can be accepted for publication in its present form.

Author Response

Response to Reviewer 1 Comments

Dear Editors, Dear Reviewer,

We were really glad to read the positive comments of Reviewer 1 regarding the revised version of our manuscript entitled: “Endogenous TOM20 proximity labeling: a Swiss-knife for the study of mitochondrial proteins in human cells” (ijms-2383907) and we would like to take this opportunity to thank once again Reviewer 1 for his constructive comments and suggestions in the previous submission.

We thank you very much in advance for considering our revised manuscript for publication.

Sincerely Yours,

S. Meurant and P. Renard.

Reviewer 2 Report (New Reviewer)

Sébastien et al. present a study on technical approaches to investigate co-translational mitochondrial protein import. The authors adress two aspects of this topic: (1) confirmation of potential reporter constructs and and their use to investigate a LARP4 knockout cell line; (2) using the TOM20-mTb fusion protein for proximity labeling of mitochondrial proteins. The techniques that were used in the study are up-to-date. One has to emphasize the fair handling of negative results and experimental difficulties. Introduction and Discussion provide a valuable overview on the adressed topics. However, many of the authors' statements are only weakly supported by the presented data due to obvious difficulties in the experimental design.

Remarks:

1. The two main aspects of the study are mixed within the Results. Section 2.1 is about reporter constructs and Section 2.3 show the use of one of them in a knockout cell line. On the other hand, Section 2.2 describes TOM20-related BioID cells lines and Sections 2.4 and 2.5 its applications. Can the authors justify this structure of the Results? Otherwise Sections 2.2 and 2.3 should be swapped.

2. Two control lines were generated to be used in combination with the TOM20-mTb cell line: TOM20-T2A-mTb and CPT1A-mTb. Due to technical problems with the first, only the second one is used as a control in proximity labeling. However, the authors are not able to demonstrate that the CPT1A-mTb fusion protein is expressed (line 229). Thus, it is difficult to interpret any of their BioID results.

3. The authors write about "high enrichment of mitochondrial proteins" in their experiment (line 563). The proportion of mitochondrial proteins within all identified proteins was 33.7% (Fig. 5B). Is this comparable with degrees of enrichment described in published studies?

4. For the constructs preALDH2-DHFR and preARG2-DHFR, the authors are not able to reproduce co-translational import that was reported in a previous study. One of the potential reasons discussed by the authors is the use of trimethoprim instead of methotrexate (lines 446-449). Whether this plays a role in assessing mitochondrial protein import could be very easily checked by performing experiments parallely with trimethoprim and methotrexate.

5. The authors use the preALDH2-DHFR construct to specifically address the functionality of the post-translation mitochondrial protein import. They state that preALDH2-DHFR is a "specific post-translational reporter". In fact, preALDH2-DHFR was reported to be imported co-translationally in a previous study while as post-translationally imported in the present study. Thus, its mode of import is contradictory.

6. In the legend to Fig. 4, both '*' and '#' are indicated as p<0.05. Is it possible that '#' actually indicates close-to-significant (p>0.05) tendency? Accordingly, is it correct to state that MED15 and GPATCH4 show significantly higher colocalization with mtHSP70 than LARP4 (lines 339-341)? What does the word 'important' mean in this context (line 339)? These results are definitelly not sufficient to claim the mitochondrial localization of the corresponding nuclear proteins.

7. The use of the term "mitochondrial transcripts" is confusing (e.g. lines 102, 139, 518, 554). There is an autonomous transcription of the mitochondrial genome within the matrix. This should not be confused with nuclear transcripts coding for mitochondrial proteins.

Author Response

Response to Reviewer 2 Comments

Dear Editors, Dear Reviewer,

It is our pleasure to re-submit our revised manuscript entitled: “Endogenous TOM20 proximity labeling: a Swiss-knife for the study of mitochondrial proteins in human cells” (ijms-2383907) for consideration as a publication in International Journal of Molecular Sciences.

We addressed the different comments and issues raised by the Reviewer 2, as detailed below. Thanks to these comments, we believe we managed to clarify the message of our study and manuscript, as well as to strengthen the conclusions we initially made. We would like to thank the Reviewer 2 for his/her valuable comments that considerably improved the structure and thus the message of the study.

We thank you very much in advance for considering our revised manuscript for publication.

Sincerely Yours,

S. Meurant and P. Renard.

Point 1: The two main aspects of the study are mixed within the Results. Section 2.1 is about reporter constructs and Section 2.3 show the use of one of them in a knockout cell line. On the other hand, Section 2.2 describes TOM20-related BioID cells lines and Sections 2.4 and 2.5 its applications. Can the authors justify this structure of the Results? Otherwise Sections 2.2 and 2.3 should be swapped.

Response 1: We totally agree with this comment, it was indeed confusing. We thus swapped the sections which makes a clearer message and we adapted the discussion accordingly.

Point 2: Two control lines were generated to be used in combination with the TOM20-mTb cell line: TOM20-T2A-mTb and CPT1A-mTb. Due to technical problems with the first, only the second one is used as a control in proximity labeling. However, the authors are not able to demonstrate that the CPT1A-mTb fusion protein is expressed (line 229). Thus, it is difficult to interpret any of their BioID results.

Response 2: We totally agree with this comment. We provide below an explanation for the non-detection of CPT1A-mTb fusion protein by western blot analysis (Figure X1), and several indirect proofs of the presence and functionality of the fusion protein.

First of all, the antibody that was used targets the C-terminal part of CPT1A, which was fused in this study to the miniTurbo protein. Therefore, we propose that the fusion may impair the ability of the antibody to recognize CPT1A.

However, three indirect arguments strongly support the expression and functionality of CPT1A-miniTurbo protein. First, a large range of biotinylated proteins were detected by western blot analysis (Supplementary figure 5D) in the CPT1A-miniTurbo cell line incubated with biotin, even though the biotinylation intensity was lower when compared to the TOM20-mTb cell line. Second, the detection of a biotinylation pattern by immunofluorescence analysis after 24 h of incubation with biotin, suggests that the construct is indeed expressed, localized at the surface of the mitochondria and is functional (Supplementary Figure 5E). Third, a careful analysis of the mass spectrometry data confirms the detection of the CPT1A-miniTurbo fusion protein. The fusion protein was indeed identified with a large sequence coverage (47 %) (FigX.1A) and the fusion peptide was even sequenced (FigX.1B), supporting the expected expression of the protein and thus the validity of the cell line.
We nonetheless agree that the biotinylation pattern of the CPT1A-mTb is lower than the one observed for the TOM20-mTb. Therefore, a normalization step is included in the mass spectrometry data analysis to take the lower abundance of proteins recovered in the CPT1A-mTb BioID into account. In addition, when selecting the candidates for functional analyses, we decided to select only highly enriched proteins to ensure their relevance and close proximity to the TOM complex.

Figure X1. MASCOT-based identification of CPT1A-mTb fusion protein in the CPT1A-mTb BioID.
A. The digested proteins were analyzed using nanoLC-ESI-MS/MS tims TOF Pro (Bruker) coupled with an UHPLC nanoElute (Bruker). The MASCOT (Matrix Science) server was used for the identification of the peptides using the HumanProteomeISO database and the sequence of the CPT1A-miniTurbo protein was added. The identified peptides matching the CPT1A-mTb are displayed and highlighted in bold red, showing the coverage of the protein. The C-terminal part of the fusion protein (residues 774 to 1045), corresponds to the sequence of the linker plus the miniTurbo biotin ligase that was fused to the CPT1A sequence. B. Sequence and mass spectrum of the fusion peptide (KGGSGDPPVATK) that was detected thanks to a trypsin miscleavage (Trypsin normally cleaves after K and R residues).

Point 3: The authors write about "high enrichment of mitochondrial proteins" in their experiment (line 563). The proportion of mitochondrial proteins within all identified proteins was 33.7% (Fig. 5B). Is this comparable with degrees of enrichment described in published studies?

Response 3:

The analysis presented in Fig.5B concerns the proteins that were identified in the BioID experiments and enriched more than two times in the TOM20-miniTurbo condition, as compared to CPT1A-mitoTurbo cell line. Both fusion proteins are localized at the surface of the mitochondria, meaning that we expected to identify mainly, in both experimental conditions, mitochondrial proteins and cytosolic proteins. Actually, the mitochondrial proteins found in common between the TOM20-mTb BioID and CPT1A-mTb BioID datasets were omitted from the 33.7 % of mitochondrial proteins. Therefore, although the figure of 33,7 % of mitochondrial proteins might seem relatively low, it is in fact quite high as it is compared to a bait also located at the OMM. Indeed, when analyzing all the proteins that were <2-fold enriched, only 6.7 % of mitochondrial proteins are found. Actually, the vast majority of mitochondrial proteins enriched in the TOM20-miniTurbo condition corresponds to proteins located inside the mitochondria (mostly in the matrix, as shown in Fig.5C), due to their biotinylation during their TOM20-dependent mitochondrial import.

In summary, we can conclude that the enrichment in mitochondrial proteins found in the TOM20-mTb condition compared to the CPT1A-mTb condition is due to the biotinylation of proteins during their import into the organelle, which thus supports the use of the TOM20-mTb to track and detect the mitochondrial proteins entry.

As demanded, we tried to compare this enrichment level to previously published studies, but this is almost impossible. The work that is the closest to experimental conditions used in our study is the one published by the group of A. Sting, who explored the proxisome of the endoplasmic reticulum surface and of the mitochondrial surface [1]. However, many parameters were fundamentally different form the current study, making it impossible to draw any comparison regarding the level of enrichment in mitochondrial proteins:

  • First, the proximity labeling of the mitochondrial surface was performed in HEK293T using an APEX2 mostly anchored in the outer mitochondrial membrane [1]. In this study, they found 84 % of mitochondrial proteins. This high proportion of mitochondrial proteins is expected as it was enriched against a cytosolic control. Our set up is fundamentally different as we consider an enrichment against another protein of the OMM.
  • Second, the selected bait protein is different: while we have targeted TOM20 as the main entry gate for mitochondrial proteins to potentially capture actors of the co-translational import, the group of A. Sting selected MAVS (Mitochondrial Anti-Viral Signaling protein). Of note, this protein is reported to have a dual subcellular localization (mitochondrial and peroxisomal).
  • Third, the authors used a really short time frame of labeling, inherent from the APEX technology, in overexpression conditions on the whole surface of the mitochondria, which differs greatly from the study found here.

Point 4: For the constructs preALDH2-DHFR and preARG2-DHFR, the authors are not able to reproduce co-translational import that was reported in a previous study. One of the potential reasons discussed by the authors is the use of trimethoprim instead of methotrexate (lines 446-449). Whether this plays a role in assessing mitochondrial protein import could be very easily checked by performing experiments parallely with trimethoprim and methotrexate.

Response 4: We totally agree with this comment and we therefore tested this hypothesis. We took the opportunity to use HeLa cells, as in the publication of Mukhopadhyay and co-workers, to also test the cell specificity hypothesis that might be another reason explaining the discrepancy observed between our study and the original one. However, as shown in FigureX2, the import feature of all the four plasmids is the same in HeLa cells as we observed previously in HCT116 cells. The use of 1 mM methotrexate (MTX), similar to the original study [2], results in identical outcomes as when cells are treated with 100 µM trimethoprim (TMP) with a post-translational import feature for preALD2-DHFR and preARG2-DHFR, a co-translational import feature for preOTC-DHFR and an intermediary phenotype for preCOX4I1-DHFR reporter (FigX.2A-D). We additionally performed the same experiment in HEK293T cells wherein we observed an identical phenotype. The quantifications of the experiment performed on HEK293T cells is shown in FigX.2E. We thus excluded the cell-type specificity hypothesis as well as the use of different DHFR inhibitors/stabilizers to explain the contradictory results obtained in this study.

In conclusion, we added an additional supplementary Figure (Supplementary Figure 2) showing the results obtained in HeLa cells for the four reporters while using TMP. We also modified the text accordingly to what is discussed hereabove:

Line 191-194: “One explanation for the discrepancy observed for the preALDH2-DHFR and preARG2-DHFR reporters might be a cell-type specificity. Indeed, while we are studying HCT116 cells, the original study used HeLa cells [45]. Therefore, the reporter plasmids were also transfected in HeLa cells (SupFig2.A-C) as well as in HEK293T (SupFig2.D). A similar feature was observed for all four reporters. The import pathway guided by the four tested pre-sequences seems thus to be conserved between different cell types.”

Line 460-466: “Several hypotheses can be proposed to explain this discrepancy. First, no information about the exact sequences that were used to generate the original reporters were available and we cannot exclude that the additional first ten amino acids of the mature protein, that were added, in this study, downstream of the pre-sequence, might have an impact. We also excluded the possibility of an effect due to the use of trimethoprim instead of methotrexate, as was used in the original study, since we could reproduce the same results using methotrexate (data not shown).  And second, micrographs in the original paper were devoid of mitochondrial labeling, making colocalization of the construct with the mitochondrial network difficult to appreciate.”

Figure X2. Characterization of the co-translational import reporters in HeLa (A-D) and HEK293T (E) cells.
A. HeLa cells were transfected 24 hours after seeding with either preALDH2-, preARG2, preCOX4I1- or preOTC-DHFR-myc plasmid. The medium was replaced 4 hours after transfection with medium supplemented with DMSO (vehicle) (A), 100 µM trimethoprim (TMP) (B) or 1 mM methotrexate (C). The next day, cells were fixed in 4 % PFA and labelled with Myc and TOM20 antibodies. Nuclei were stained using DAPI. Micrographs were acquired on a Zeiss LSM900 Airyscan confocal microscope. Scale bar = 10 µm. D. The quantifications were performed using Fiji software with the ComDet colocalization plugin. Analyses were done on 5-10 cells in one biological replicate. E. Quantifications of the data obtained on HEK293T cells that followed the same treatments and analyses. Analyses were performed on 10-20 cells in one biological replicate.

Point 5: The authors use the preALDH2-DHFR construct to specifically address the functionality of the post-translation mitochondrial protein import. They state that preALDH2-DHFR is a "specific post-translational reporter". In fact, preALDH2-DHFR was reported to be imported co-translationally in a previous study while as post-translationally imported in the present study. Thus, its mode of import is contradictory.

Response 5: It is true that our data is contradictory with the work of Mukhopadhyay et al. (2004). However, careful analysis reveals several questionable aspects and flaws in this publication, such as the absence of loading controls in western blots, the absence of replicates and thus of statistical analyses, the use of epi-fluorescence microscopy and not confocal microscopy, the assumption of a subcellular localization of a reporter protein without any organelle markers, etc. In addition, the demonstration that the reporter proteins are co-translationally imported is only based on differences in the electrophoresis pattern between the reporter proteins that reveal the mature and the processed form of the reporter protein. In some figures, the reporter proteins with or without the preALDH2 pre-sequence have a different electrophoretic pattern (Fig5A) while, in other figures, the apparent molecular weight is the same (Fig3, lanes b-c and Fig4 lanes b-c), a point that is not even discussed by the authors. But most importantly, critical information is missing in the publication of Mukhopadhyay et al. (2004) that would allow us to identify and/or discuss the origin of the discrepancy between their data and our results. For example, the exact sequences and length of the signal peptides studied in this previous publication are not provided, and we cannot exclude that the 10 first amino acids of the mature protein that we have added to the pre-sequence (Supplementary Figure 1) do not influence the import pathway of the reporter DHFR protein.

Our results are controlled with the appropriate organelle markers, are properly quantified, are statistically significant (n≥ 3 for each experiment), and are reproducible in three different human cell types/lines. Moreover, we were able to demonstrate the co-translational feature of the pre-OTC-DHFR reporter, suggesting the approach and the read-outs we used are appropriate. Therefore, we are totally confident about the post-translational feature of the preALDH2-DHFR construct described in Supplementary Figure 1A, and we are convinced that it can be used confidently to assess post-translational import. However, we agree with the Reviewer that the choice of the “specific” word in this context might be confusing and we thus removed it (line 230).

Point 6: In the legend to Fig. 4, both '*' and '#' are indicated as p<0.05. Is it possible that '#' actually indicates close-to-significant (p>0.05) tendency? Accordingly, is it correct to state that MED15 and GPATCH4 show significantly higher colocalization with mtHSP70 than LARP4 (lines 339-341)? What does the word 'important' mean in this context (line 339)? These results are definitelly not sufficient to claim the mitochondrial localization of the corresponding nuclear proteins.

Response 6: The two symbols used in the Fig.4 correspond to two kinds of comparisons with either the comparison with AKAP1 (*) or LARP4 (#), used as the controls, and correspond thus to two independent statistical analyses but both symbols correspond really to significancy with p<0.05. In this context, we think it is correct to state a higher colocalization with mtHSP70 for MED15 and GPATCH4 when compared to LARP4. We nonetheless agree that the word “important” was not the most appropriate and we thus changed in the text as follow:

Line 355-358: “In addition, while the colocalization of the three proteins with mtHSP70 was not as strong as for the AKAP1 positive control, it was higher than the colocalization of LARP4 with mtHSP70, and significantly different for MED15 and GPATCH4 (Fig4.C).”

In order to confirm the mitochondrial localization of the three nuclear proteins, we therefore performed a cell fractionation to look for the presence of the candidates in the mitochondrial fraction, as shown in the Figure X3. Interestingly, we could specifically enrich MED15 in the mitochondrial fraction. However, CPSF2 could not be detected due to technical limitation related to the antibody used and GPATCH4 was not detected in the mitochondrial fraction. Additional optimization of the fractionation should be done to obtain a publishable figure since the mitochondrial contamination (shown by mtHSP70 immunodetection) in the nuclear fraction is still high. However, only a low nuclear contamination (mitochondrial lamin A/C signal corresponding to only 13.8 % of its nuclear signal) is observed in the mitochondrial fraction wherein MED15 could be clearly detected (mitochondrial signal corresponding to 59.3 % of its nuclear signal) (Fig.X3).
Additionally, a small molecular weight shift is observed for all the proteins found in the mitochondrial fraction, probably due to the different densities of the different samples loaded on the gel, containing slight changes in sucrose concentration. Interestingly, at least three bands can be immuno-detected with the MED15 antibodies, and the electrophoretic pattern seems to be cell compartment-specific. These different bands might correspond to various isoforms of the protein, that have been described in yeast [3]. In addition, different post-translational modifications have been predicted (Q96RN5, UniProt) or demonstrated [4] for MED15, that could also contribute to this potentially interesting compartment-specific electrophoretic pattern. The mitochondria-specific isoform could be further investigated with the immuno-purification of the mitochondrial MED15 and de novo sequencing by mass spectrometry.
Regarding GPATCH4 results, the absence of detection for the protein in the mitochondrial fraction could suggest a loose/weak? and labile interaction between the protein and the outer mitochondrial membrane and would not be found inside the organelle. This hypothesis is further supported by the high enrichment of the protein in the cytosolic fraction, in line with the observation that the protein is also likely to be found outside the nucleus.

In conclusion, the set of data presented below (FigX.3) confirms our results, at least for MED15, and supports the message delivered in this study with the possibility to use the TOM20-mTb cell line for the detection of new mitochondrial proteins, expanding the list of those proteins. Altogether, data from different types of experiments (BioID, immunofluorescence and cell fractionation) support our claim of a mitochondrial localization (inside or at the surface of the mitochondria) of at least one of our nuclear candidates.

Figure X3. Mitochondrial fractionation supports the mitochondrial localization of MED15.
HEK293T cells were seeded and scraped in 0.2 M sucrose 24 hours after. The cells were then homogenized using a dounce and 5 % of the homogenate were saved (H 5 %) whereas the rest was centrifuged for 10 minutes at 4°C at 1000 x g. The nuclear pellet was resuspended in RIPA lysis buffer (N fraction) and the supernatant was then ultracentrifuged for 2 min at 4°C at 6587 x g. The mitochondrial pellet (M fraction) was resuspended in RIPA lysis buffer and the supernatant was recovered as the cytosolic fraction for which 1.5 % was saved (C 1.5 %)An equivalent of 5 % of the nuclear and mitochondrial fraction proteins were resolved by SDS-PAGE, respectively. The abundance of MED15, lamin A/C (nuclear marker) (#612162, BD Biosciences), mtHSP70 (mitochondrial marker) and GPATCH4 were analyzed by western blot (n = 3). The fluorescent signals of MED15 and lamin A/C in the N and M fractions were measured and the fluorescence intensity of signal in the M fraction was expressed as a percentage of the fluorescence signal found in the N fraction, for both proteins. The 13,8 % calculated for the lamin A/C signal thus corresponds to the nuclear contamination in the mitochondrial fraction.

Point 7: The use of the term "mitochondrial transcripts" is confusing (e.g. lines 102, 139, 518, 554). There is an autonomous transcription of the mitochondrial genome within the matrix. This should not be confused with nuclear transcripts coding for mitochondrial proteins.

Response 7: We agree with this comment, it is indeed confusing regarding the dual genetic origin of the mitochondrial transcripts. We thus modified the manuscript accordingly with the following specification: “nucleus-encoded mitochondrial transcripts” at the lines: 99, 102, 105, 110, 113, 117, 124, 126, 134, 141, 490, 514, 567, 583.

References:

  1. Hung, V.; Lam, S. S.; Udeshi, N. D.; Svinkina, T.; Guzman, G.; Mootha, V. K.; Carr, S. A.; Ting, A. Y. Proteomic Mapping of Cytosol-Facing Outer Mitochondrial and ER Membranes in Living Human Cells by Proximity Biotinylation. Elife 2017, 6. https://doi.org/10.7554/eLife.24463.
  2. Mukhopadhyay, A.; Ni, L.; Weiner, H. A Co-Translational Model to Explain the in Vivo Import of Proteins into HeLa Cell Mitochondria. Biochem. J. 2004, 382 (1), 385–392. https://doi.org/10.1042/BJ20040065.
  3. Gallagher, J. E. G.; Ser, S. L.; Ayers, M. C.; Nassif, C.; Pupo, A. The Polymorphic PolyQ Tail Protein of the Mediator Complex, Med15, Regulates the Variable Response to Diverse Stresses. Int. J. Mol. Sci. 2020, Vol. 21, Page 1894 2020, 21 (5), 1894. https://doi.org/10.3390/IJMS21051894.
  4. Ishikawa, H.; Tachikawa, H.; Miura, Y.; Takahashi, N. TRIM11 Binds to and Destabilizes a Key Component of the Activator-Mediated Cofactor Complex (ARC105) through the Ubiquitin-Proteasome System. FEBS Lett. 2006, 580 (20), 4784–4792. https://doi.org/10.1016/J.FEBSLET.2006.07.066.

Round 2

Reviewer 2 Report (New Reviewer)

The authors addressed all my concerns and performed additional experiments (unfortunately, most of the new results were not integrated into the main manuscript or the supplement). Nevertheless, the manuscript can be accepted in its present form.

This manuscript is a resubmission of an earlier submission. The following is a list of the peer review reports and author responses from that submission.

Round 1

Reviewer 1 Report

In their manuscript, Sébastien and colleagues performed a proximity labelling approach to study the wider interactome/proxisome of the outer mitochondrial membrane protein TOM20 with the aim to analyze mitochondrial co-translational import mechanisms. The manuscript is well-written, featuring a good representation and explanation of the obtained results as well as a thorough - if not even too lengthy - discussion. Despite failing in identifying specific effectors related to this import mechanisms and, thereby, providing a further proof of the respective hypothesis, the methodology-oriented work is well-executed and of interest for a readership focusing on this specific field of research. However, textually, the manuscript focuses much on mitochondrial co-translational import, while not delivering any usable and supportive findings in this context. Consequently, changing the textual focus from mitochondrial co-translational import to the technical aspects of the proximity labelling method would rather live up to the provided dataset and stress the versatility of the technique than overemphasizing the failed experimental outcomes. In sum, the study gives a good additional insight in the capabilities of proximity labelling-based proteomic analysis for a better understanding of mitochondrial processes at the mitochondrial membrane, while lacking evidence for its suitability in assessing mitochondrial co-translational import.

A few major and minor points need to be addressed before the manuscript is suitable for publication:

Previous studies have successfully engaged in analyzing the proxisome of TOM20 using biotin-based proximity labelling approaches in different models by employing BirA, APEX or even split-BioID/TurboID pairs. References to these works, specifically Kwak et al., 2020 (doi.org/10.1073/pnas.1916584117), Cho et al., 2020 (doi.org/10.1073/pnas.1919528117), Vardi-Oknin and Arava, 2019 (doi.org/10.3389/fcell.2019.00305), and Lee et al., 2016 (doi.org/10.1016/j.celrep.2016.04.064), have been omitted in this manuscript. Authors should include them and discuss their results in the light of these earlier studies. This is specifically necessary in the context of the publication by Vardi-Oknin and Arava, who claim to have provided certain hints towards co-translational import into mitochondria using a TOM20-BirA fusion protein.

Page 2: The graphical abstract is slightly disproportional regarding the displayed elements. Reducing the size of the mitochondria and increasing the sizes of the elements at the bottom would improve the illustration. Adjustments should also be done for the labeling of the subunits of the TOM complex and the surrounding proteins, which are hard to read.

Results 2.1: The utilized reporter constructs consist of the MTS and 10 amino acids of the four analyzed proteins + N-term ecDHFR. While the results are convincing, is it possible that another motif downstream of the enzymes’ protein sequence might contribute to the co-translational import signaling? Would the experiments have a different outcome if fusion proteins of the entire enzyme + N-term ecDHFR were generated? Could the authors please comment?

Fig. 1: Can the authors include a schematic representation of the utilized reporter constructs, perhaps also showing the respective sequences of the analyzed enzymes?

Results 2.2: As the authors have failed to generate a reliable control for cytoplasmic mTb using the TOM20Cter-T2A-mTb construct, have they considered (randomly) knocking in mTb alone as a respective control instead?

Suppl. Fig. 1A: Authors should provide a western blot analysis to confirm the expression of the respectively generated fusion proteins, perhaps by using the mTb-specific antibodies or antibodies against TOM20 and CPT1A. Thereby, the success rate of the knock-in as well as the ribosomal skipping for the T2A-based strategy could have been assessed.

Suppl. Fig. S2B: The table in S2B should be transformed in a Suppl. Table. Moreover, introduce abbreviations “FC” and “GO” in the respective legend.

I have unsuccessfully tried to access the raw mass spectrometry data on the PRIDE database website using the identifier PXD038821. Have the datasets been made available yet?

Line 237: “FigureS=3.A” (remove the “=”)

Result 2.3: As knockout of LARP4 did not lead to desired effects regarding mitochondrial import, did the authors try to analyze cells with a AKAP1 knockout or knockdown? Moreover, did the authors analyze the effect on endogenously expressed OTC instead of using the preOTC-DHFR-myc construct?

Suppl. Fig. S5: Could the authors provide a merge/overlay image for the micrographs in A? Font sizes can be adjusted for the axis-labeling in B, C, D.

Fig. 5D, E: Color/symbol legend and labeling are too small and hard-to-read.

Results 2.4 and 2.5: The findings on the three RNA-binding proteins as parts of the proxisome of TOM20 as well as the mitochondrial enrichment data would be more convincing if the authors provided additional validation experiments for a selection of proteins of interest, for example using mitochondrial fractionation assays and western blot detection of the specific candidates. Without further validation of at least some targets, the reproducibility and validity of the discussed results will remain contestable.

The findings on the correlation between mitochondrial enrichment and the protein half-life are interesting. However, it adds another aspect to the entire study which does not ideally blend in with the residual dataset of the manuscript and, as it does only appear as supplementary data without any further experimental validation, could be left out.

Suppl. Fig. S6B: Authors should select and indicate a few exemplary proteins of interest in their graph.  

Line 391/392: change “the authors” to “we”

Line 475: change “larp protein” to “LARP4” or “LARP”, as the “p” in LARP already stands for “protein”

Minor general comments:

Authors should doublecheck for further lowercase/uppercase, and formatting inconsistencies, e.g., Fig. 1 legend, “24 hours” or “4h” (decision for one format; if second, space missing), Suppl. Figs. S1 figure legend, “D. Wild-Type” (“type” in lowercase), S2B column headers “log2(FC)” (“2” in subscript), “Pvalue” (missing space), S6B, y-axis “Protein Half-life” (lower case “half”); line 71: “…, 3 proteins…” (change to “three protein”), see also lines 138/139 “Only 3 enzymes…” (change to “three enzymes”), or Figure S1: “…of the 3 BioID cell …” (change to “three BioID”), etc.

Lines 59/72/81/243/473: Lowercase/uppercase inconsistency for Drosophila

When p-values are given in the figures, spaces are missing.

Author Response

Response to Reviewer 1 Comments

Dear Editors, Dear Reviewer,

It is our pleasure to re-submit our revised manuscript entitled: “Endogenous TOM20 proximity labeling: a Swiss-knife for the study of mitochondrial proteins in human cells” (ijms-2129480) for consideration as a publication in International Journal of Molecular Sciences.

We addressed the different comments and issues raised by the reviewers, as detailed in the comments below. Thanks to these comments, we believe we managed to clarify the message of our study and manuscript, as well as to strengthen the conclusions we initially made.

We thank you very much in advance for considering our revised manuscript for publication.

Sincerely Yours,

  1. Meurant and P. Renard.

Point 1: In their manuscript, Sébastien and colleagues performed a proximity labelling approach to study the wider interactome/proxisome of the outer mitochondrial membrane protein TOM20 with the aim to analyze mitochondrial co-translational import mechanisms. The manuscript is well-written, featuring a good representation and explanation of the obtained results as well as a thorough - if not even too lengthy - discussion. Despite failing in identifying specific effectors related to this import mechanisms and, thereby, providing a further proof of the respective hypothesis, the methodology-oriented work is well-executed and of interest for a readership focusing on this specific field of research. However, textually, the manuscript focuses much on mitochondrial co-translational import, while not delivering any usable and supportive findings in this context. Consequently, changing the textual focus from mitochondrial co-translational import to the technical aspects of the proximity labelling method would rather live up to the provided dataset and stress the versatility of the technique than overemphasizing the failed experimental outcomes. In sum, the study gives a good additional insight in the capabilities of proximity labelling-based proteomic analysis for a better understanding of mitochondrial processes at the mitochondrial membrane, while lacking evidence for its suitability in assessing mitochondrial co-translational import.

Response 1: First of all, we would like to thank the reviewer 1 for his precious comments thanks to which the manuscript was substantially improved.

We changed the textual focus stressing more now on the proximity labelling technique rather than mitochondrial co-translational import. The message of the paper now emphasizes on the versatility of use of proximity labelling for mitochondrial proteins study. We propose the use of endogenous TOM20 proximity labelling as a new tool for mitochondrial processes study such as co-translational import process and propose a strategy for its high-throughput unbiased study.

The whole abstract and introduction were thus modified accordingly.

Point 2: Previous studies have successfully engaged in analyzing the proxisome of TOM20 using biotin-based proximity labelling approaches in different models by employing BirA, APEX or even split-BioID/TurboID pairs. References to these works, specifically Kwak et al., 2020 (doi.org/10.1073/pnas.1916584117), Cho et al., 2020 (doi.org/10.1073/pnas.1919528117), Vardi-Oknin and Arava, 2019 (doi.org/10.3389/fcell.2019.00305), and Lee et al., 2016 (doi.org/10.1016/j.celrep.2016.04.064), have been omitted in this manuscript. Authors should include them and discuss their results in the light of these earlier studies. This is specifically necessary in the context of the publication by Vardi-Oknin and Arava, who claim to have provided certain hints towards co-translational import into mitochondria using a TOM20-BirA fusion protein.

Response 2: We apologize for the missing references that are now included into the present manuscript as such:

Page 3: “The generation and development of split versions of the biotin ligases also enable to go further and to identify specific factors of organelles contact-sites with, for example, the identification of Mitochondria-Associated Membranes (MAM) proteome [1,2].”

Page 4: “Third, Clueless (Clu) protein, located at the OMM, associates with PINK1-Parkin [3], and binds both mitochondria transcripts and ribosomes and is therefore proposed to mediate mitochondrial co-translational import [4].”

“Interestingly, a more recent study identified the mitochondrial transcripts undergoing translation in the vicinity of the TOM complex in human cells by taking advantage of the elegant proximity-specific ribosome profiling technique [5,6]. In this study, the authors highlighted the role of CLUstered mitochondria protein Homolog (CLUH), the mammalian ortholog of Clu, in the regulation of the mitochondrial localized translation of several mitochondrial transcripts, showing a potential involvement in the mitochondrial co-translational import process [5].”

In addition, we now discuss about the recent work done by Vardi-Oknin and Arava and put our results in perspective regarding the results obtained in their study.

Page 16: “While Vardi-Oknin and Arava detected CLUH using proximity-specific ribosome profiling, we were unable to detect the protein in our BioID analyses. This may be explained by the cell type used in our present study (HCT116), different from the one used in Vardi-Oknin and Arava’s study (HEK293T) [5]. Alternatively, more recent results seem to indicate that CLUH would be mainly located in stress granule-like structures where it would exert its translational regulator function [7]. Therefore, even though this protein regulates the translation of mitochondrial transcripts, it would not be involved in the regulation of their co-translational import. This is in line with the results obtained here and the apparent absence of CLUH at the surface of the mitochondria.”

Point 3: Page 2: The graphical abstract is slightly disproportional regarding the displayed elements. Reducing the size of the mitochondria and increasing the sizes of the elements at the bottom would improve the illustration. Adjustments should also be done for the labeling of the subunits of the TOM complex and the surrounding proteins, which are hard to read.

Response 3: Thank you for the comment. We improved the graphical abstract which is now easier to read.

Point 4: Results 2.1: The utilized reporter constructs consist of the MTS and 10 amino acids of the four analyzed proteins + N-term ecDHFR. While the results are convincing, is it possible that another motif downstream of the enzymes’ protein sequence might contribute to the co-translational import signaling? Would the experiments have a different outcome if fusion proteins of the entire enzyme + N-term ecDHFR were generated? Could the authors please comment?

Response 4: It may indeed be the case that a co-translational import sequence is located in another part of the mitochondrial protein used as a reporter but no co-translational sequence has been described so far. Indeed, only the addressing sequences, which will dictate the final submitochondrial localization, are well described so far [8–10] but nothing is known on co-translational import specific cis-elements. However, the first reason why we decided to work with the MTS (Mitochondrial Targeting Sequence) + 10 amino acids is to reproduce what was done in the work of Mukhopadhyay and colleagues, which was the first and only evidence of human mitochondrial proteins that are co-translationally imported [11]. In addition, the mitochondrial co-translational import has been demonstrated, in yeast, to rely heavily on TOM20 thanks to its association with the nascent-chain peptide, through the binding of the MTS [8,12,13]. Indeed, the MTS of mitochondrial proteins has been identified, in yeast, to be one of the major elements ensuring the mitochondrial co-translational import process along with the TOM20 protein, the translation machinery and specific RNA-binding proteins such as PUF3 proteins [12]. For those reasons, we selected the pre-sequence of the selected mitochondrial proteins and the first 10 amino acids were also initially added in case the beginning of the protein was also important [11]. Nonetheless, other parts of the proteins might still be important/essential for the co-translational import process and the fusion of the whole proteins could result in a different outcome and should be tested.

This was implemented in the discussion as such:

Page 14: “Importantly, the different reporters used in our study are all based on the MTS of the mitochondrial proteins which can be a limitation since the MTS is probably not the only sequence triggering and promoting the co-translational import of the corresponding protein. Indeed, other sequences of the proteins may still be important/essential for the co-translational import process. Therefore, constructs based on the fusion of the full-length proteins could result in a different and more relevant outcome and should be tested.”

Point 5: Fig. 1: Can the authors include a schematic representation of the utilized reporter constructs, perhaps also showing the respective sequences of the analyzed enzymes?

Response 5: We added a supplementary figure (Figure S1, page 22) with schematic representation of the four reporter constructs with their specific respective sequences. Therefore, all the numbers of the supplementary figures were shifted of one number up.

Point 6: Results 2.2: As the authors have failed to generate a reliable control for cytoplasmic mTb using the TOM20Cter-T2A-mTb construct, have they considered (randomly) knocking in mTb alone as a respective control instead?

Response 6: The problem here would be the site of insertion of the randomly knocked-in miniTurbo transgene. The elegance of the initial control (TOM20Cter-T2A-mTb) was that the cytosolic mTb would have been expressed at the same level as the TOM20-mTb construct. Therefore, to obtain a similar control, it would require the knock-in of the mTb gene under the control of the TOM20 promoter which would then be found in three to four copies in the cell genome, which could affect the transcription of the endogenous TOM20 alleles. Alternatively, integrating the gene encoding mTb randomly in the genome would not allow to control the level of expression of this transgene, in addition to potentially dysregulate the expression of endogenes at the integration locus. Another possibility would have been to target a privileged genomic integration site with no impact on the expression of coding or non-coding elements, such as the Rosa26 locus in mouse, but it is less clear whether such sites can be found in human cells or not. In addition, obtaining the same expression level as the endogenous TOM20 would still represent a major challenge with this strategy. 

In the current work, the best solution would be first to obtain a good antibody recognizing specifically the miniTurbo protein lo assess its expression in the cell line used in our study. We could then check whether the absence of cytosolic biotinylation is due to a really low abundance of the protein or to a defective biotinylating activity of the enzyme. Finally, we could also use another self-cleavable linker, such as EA2 or P2A, or linkers in tandem to see whether we obtain better results as it has already been demonstrated using GFP-based constructs [14].

Indeed, in another study using T2A-GFP constructs in HEK293T, these authors reported a reduced protein expression, by 30-50 %, downstream of the T2A sequence which might explain the low cytosolic biotinylating activity and the absence of detection of the cytosolic mTb observed here [14].

Point 7: Suppl. Fig. 1A: Authors should provide a western blot analysis to confirm the expression of the respectively generated fusion proteins, perhaps by using the mTb-specific antibodies or antibodies against TOM20 and CPT1A. Thereby, the success rate of the knock-in as well as the ribosomal skipping for the T2A-based strategy could have been assessed.

Response 7: We implemented as the Figure S2B the required western blot that we comment as such:

Page 7: “Following the generation of the three BioID cell lines, the expression of all three fusion proteins was monitored by western blotting, after different times of incubation with biotin to ensure that the biotinylating activity does not modify the stability of the fusion protein over time (SupFig2.B). For each cell line, both wild type and fused forms of the proteins could be detected, as expected for CRISPR-Cas9 mediated knock-in, yielding mainly heterozygous cell lines. The use of the commercially available BirA* antibody didn’t allow the detection of the miniTb variant. Nonetheless, for the TOM20-T2A-mTb cell line, both cleaved and complete constructs were detected with the TOM20 antibody. For the TOM20-mTb cell line, the fusion protein seemed to be expressed at the same level as the wild type allele validating the endogenous expression of the construct. Unfortunately, the CPT1A-mTb protein could not be detected but the abundance of the wild type protein seemed to be lower (by about 50 %) suggesting either a reduced expression of the fusion protein or a reduced capacity of the antibody to recognize the modified protein (SupFig2.B). Since the fusion of the mTb protein could still potentially affect the endogenous function of the fused proteins, the use of heterozygous cell lines could be an advantage to reduce this potential effect on both mitochondria and cell physiology.”

Page 15: “However, a reduced protein expression downstream of the T2A sequence has been reported using T2A-GFP constructs in HEK293T and might explain the really low cytosolic biotinylating activity of this construct [14].”

We additionally changed the legend of the Figure S2 as it follows:

Page 23: “C. T2A-miniTurbo (T2A-mTb), TOM20-miniTurbo (TOM20-mTb) and CPT1A-miniTurbo (CPT1A-mTb) HCT116 cells were seeded and 50 µM biotin was added 24 hours after for 0, 30 minutes, 2 or 24 hours. Cells were harvested and lysed in RIPA buffer and 20 µg of proteins were resolved by SDS-PAGE. CPT1A and TOM20 (B) were revealed as well as biotinylated proteins (C), tubulin was used as a loading control (n = 1). D. T2A-mTb, TOM20-mTb and CPT1A-mTb cells were incubated for 24 hours in the presence of 50 µM biotin 24 hours after seeding. Cells were then harvested and lysed in RIPA buffer. A maximum amount of proteins (2.3 mg) was loaded on streptavidin beads and incubated on wheel for 24 hours at 4 °C. The beads were then rinsed and resuspended prior to trypsin digestion and mass spectrometry analysis using a timsTOF (Bruker). An equivalent of 2.5 % of total proteins (57.5 µg) was loaded on acrylamide gel as input along with 2.5 % of the pull-down (PD) for each experimental condition. Biotinylated proteins and tubulin were revealed using specific antibodies (n = 3).  E. Wild type (WT), T2A-mTb, TOM20-mTb and CPT1A-mTb HCT116 were incubated for 24 hours in the presence of 50 µM biotin, 24 hours after seeding.”

Point 8: Suppl. Fig. S2B: The table in S2B should be transformed in a Suppl. Table. Moreover, introduce abbreviations “FC” and “GO” in the respective legend.

I have unsuccessfully tried to access the raw mass spectrometry data on the PRIDE database website using the identifier PXD038821. Have the datasets been made available yet?

Response 8: The supplementary table was generated and the abbreviations were introduced in the respective legend as follows:

Page 24: Legends: “Figure S3: Complementary GO term analysis reveals enrichment of translation-related biological processes.  

Gene Ontology (GO) enrichment analysis done on the ≥ 2-fold-enriched proteins in the TOM20-mTb condition relative to CPT1A-mTb, using DAVID platform [54], showing the top enriched Biological Process GO terms. Translation-related GO terms are highlighted.”

Table S1: Enrichment of translation-related proteins in the BioID dataset.
Table showing all the ≥ 2-fold-enriched proteins and their associated p value displaying Gene Ontology (GO) terms related to either TOM complex, translation initiation activity or ribosomes. FC: Fold Change.”

Regarding the access to the mass spectrometry data, it is not officially available yet but you can access the data as a reviewer after having logged in using the user name: reviewer_pxd038821@ebi.ac.uk and the password: p59LtaI8.

Point 9: Line 237: “FigureS=3.A” (remove the “=”).

Response 9: The “FigureS=3.A” was probably generated during the pdf formatting since we don’t observe this typing fault in the original manuscript.

Point 10: Result 2.3: As knockout of LARP4 did not lead to desired effects regarding mitochondrial import, did the authors try to analyze cells with a AKAP1 knockout or knockdown? Moreover, did the authors analyze the effect on endogenously expressed OTC instead of using the preOTC-DHFR-myc construct?

Response 10: The knock-down (KD) of AKAP1, first assessed by western blot (Fig.X1A), showed similar results when compared to the knock-out of LARP4, with no difference in the localization of the preOTC-DHFR co-translational reporter, even in the presence of trimethoprim (TMP) (FigX1.B). Indeed, no difference in the distribution pattern of the reporter could be observed following the KD of AKAP1, whereas the strong KD of the protein was confirmed in the transfected cells by immunodetection of the AKAP1 protein. Therefore, the transient invalidation of AKAP1 doesn’t seem to affect the co-translational import of the product of the specific construct, further supporting the results presented in the manuscript. Interestingly, the use of a transient invalidation approach instead of a stable one, as used before, allows to minimize potential compensatory mechanisms that may arise following constitutive invalidation.

Regarding the localization and abundance of the endogenous OTC protein, we couldn’t make any observation due to the poor specificity of the used antibody (#26470-1-AP, Proteintech). Therefore, we decided to detect another additional endogenous protein in both LARP4 WT and KO conditions. We thus choose to immunodetect the fumarase, an enzyme that has been previously described to be obligatory co-translationally imported, at least in yeast [15]. However, no change in either the localization nor in the abundance of the protein could be observed in the KO condition when compared to WT cells with a maintained mitochondrial localization as demonstrated with the colocalization with the mtHSP70 matrixial marker (FigX1.C). However, the co-translational import of the human fumarase should first be addressed and confirmed before using this protein as a new co-translational import marker.

Figure X1. Absence of effect of AKAP1 KD and LARP4 KO on the assessed mitochondrial co-translational import markers.
A.
HEK293T cells were seeded and transfected 24 hours after with either 20 µM Non-Targeting silencing RNA (NT siRNA) (#D-001810-10-20, Dharmacon) or 20 µM AKAP1 SMARTPOOL siRNAs (#L-011426-00-0005, Dharmacon). After 48 hours, the cells were harvested and lysed in RIPA buffer and 20 µg of proteins was resolved by SDS-PAGE. AKAP1 and tubulin were analyzed by western blot (n = 1). B. HEK293T cells were seeded and transfected 24 hours after with either 20 µM NT siRNA or 20 µM AKAP1 SMARTPOOL siRNAs. The cells were then transfected 24 hours after with the preOTC-DHFR-myc reporter. The medium was replaced 4 hours after transfection with medium supplemented (in the knock-down (KD) condition) or not (in the NT condition) with 100 µM trimethoprim (TMP). 24 hours after, the cells were fixed in 4 % PFA and labelled with Myc and AKAP1 (#5203, Cell Signaling Technology) antibodies. Nuclei were stained using DAPI. Micrographs were acquired on a Zeiss LSM900 Airyscan confocal microscope (n = 1). Scale bar = 10 µm. C. WT and LARP4 KO HEK293T cells were seeded, fixed 24 hours after and labelled with Fumarase (#PA5-22064, ThermoFisher Scientific) and mtHSP70 antibodies. Nuclei were stained using DAPI. Micrographs were acquired on a Zeiss LSM900 Airyscan confocal microscope (n = 4). Scale bar = 10 µm.

Point 11: Suppl. Fig. S5: Could the authors provide a merge/overlay image for the micrographs in A? Font sizes can be adjusted for the axis-labeling in B, C, D.

Response 11: We modified the old Figure S5, now the Figure S6, by adding the merge micrographs and we adjusted all font of axis-labeling.

Point 12: Fig. 5D, E: Color/symbol legend and labeling are too small and hard-to-read.

Response 12: We tried as best as we could to make the graphs bigger and easier to read by changing the setting of the Figure 5.

Point 13: Results 2.4 and 2.5: The findings on the three RNA-binding proteins as parts of the proxisome of TOM20 as well as the mitochondrial enrichment data would be more convincing if the authors provided additional validation experiments for a selection of proteins of interest, for example using mitochondrial fractionation assays and western blot detection of the specific candidates. Without further validation of at least some targets, the reproducibility and validity of the discussed results will remain contestable.

Response 13: We totally agree with this comment and we thus performed a mitochondria fractionation, as shown in the Figure X2. Interestingly, we could enrich specifically MED15 in the mitochondrial fraction. However, CPSF2 could not be detected due to technical limitation related to the antibody used and GPATCH4 was not detected in the mitochondrial fraction. Additional optimization of the fractionation should be done to obtain a publishable figure since the mitochondrial contamination (shown by mtHSP70 immunodetection) in the nuclear fraction is still high. However, only a low nuclear contamination (mitochondrial lamin A/C signal corresponding to only 13.8 % of its nuclear signal) is observed in the mitochondrial fraction wherein MED15 could be more importantly detected (mitochondrial signal corresponding to 59.3 % of its nuclear signal) (Fig.X2).
Additionally, a small molecular weight shift is observed for all the proteins found in the mitochondrial fraction, probably due to the different densities of the different samples loaded on the gel, containing more or less sucrose. Interestingly, at least three bands can be immuno-detected with the MED15 antibodies, and the electrophoretic pattern seems to be cell compartment-specific. These different bands might correspond to various isoforms of the protein, that have been described in yeast [16]. In addition, different post-translational modifications have been predicted (Q96RN5, UniProt) or demonstrated [17] for MED15, that could also contribute to this potentially interesting compartment-specific electrophoretic pattern. The mitochondria-specific isoform could be further investigated with the immuno-purification of the mitochondrial MED15 and de novo sequencing by mass spectrometry.
Regarding GPATCH4 results, the absence of detection for the protein in the mitochondrial fraction could suggest a loose and labile interaction between the protein and the outer mitochondrial membrane and would not be found inside the organelle. This hypothesis is further supported by the high enrichment of the protein in the cytosolic fraction, in line with the observation that the protein is likely found outside the nucleus too

In conclusion, the data presented below (FigX.2) confirm our results, at least for MED15, and support the message delivered in this study with the possibility to use the TOM20-mTb cell line for the detection of new mitochondrial proteins, expanding the list of those proteins.

Figure X2. Mitochondrial fractionation supports the mitochondrial localization of MED15.
HEK293T cells were seeded and scraped in 0.2 M sucrose 24 hours after. The cells were then homogenized using a dounce and 5 % of the homogenate was saved (H 5 %) whereas the rest was centrifuged for 10 minutes at 4°C at 1000 xg. The nuclear pellet was resuspended in RIPA lysis buffer (N fraction) and the supernatant was then ultracentrifuged for 2 min at 4°C at 6587 xg. The mitochondrial pellet (M fraction) was resuspended in RIPA lysis buffer and the supernatant was recovered as the cytosolic fraction for which 1.5 % was saved (C 1.5 %)An equivalent of 5 and 10 % of the nuclear and mitochondrial fraction proteins were resolved by SDS-PAGE, respectively. The abundance of MED15, lamin A/C (nuclear marker) (#612162, BD Biosciences), mtHSP70 (mitochondrial marker) and GPATCH4 were analyzed by western blot (n = 3). The fluorescent signals of MED15 and lamin A/C in the N and M fractions were measured and the fluorescence intensity of signal in the M fraction was expressed as a percentage of the fluorescence signal found in the N fraction, for both proteins. The 13,8 % figure for the lamin A/C signal thus corresponds to the nuclear contamination in the mitochondrial fraction.

Point 14: The findings on the correlation between mitochondrial enrichment and the protein half-life are interesting. However, it adds another aspect to the entire study which does not ideally blend in with the residual dataset of the manuscript and, as it does only appear as supplementary data without any further experimental validation, could be left out.

Response 14: Since the scope of the paper is now more focused on the multiple use of an endogenous TOM20-based proximity labeling cell line for the study of mitochondrial proteins, we estimated that this potential application for the prediction of matrixial protein half-life is of importance, and we maintained this piece of data in the manuscript.

Point 15: Suppl. Fig. S6B: Authors should select and indicate a few exemplary proteins of interest in their graph.

Response 15: We further highlighted five proteins corresponding to the only proteins with available half-life data in HCT116 [18]. The legend of figure S7B was consequently modified as follows:

Page 28: “Only few half-life data for HCT116 mitochondrial matrix proteins, highlighted on the graph in red, are available [18] and, thus, additional half-life data from human primary hepatocytes [19] were added to the analysis.”

Point 16: Line 391/392: change “the authors” to “we”.

Response 16: There we meant to speak about the first paper identifying OTC, ALDH2 and ARG2 as co-translationally imported proteins [11] and not of our own study. We changed in the text to make it clearer as such:

Page 14: “In this pioneered study, the authors used DHFR-based reporters stabilized or not with methotrexate and described the co-translational import of the three reporters using western blot and immunofluorescence analyses.”

Point 17: Line 475: change “larp protein” to “LARP4” or “LARP”, as the “p” in LARP already stands for “protein”.

Response 17: We removed the word “protein” which was indeed redundant, as follows:

Page 15: “Similarly, the Drosophila ortholog of AKAP1, MDI, was also described to mediate localized translation of mitochondrial transcripts at the surface of mitochondria, in association with Larp, and the authors proposed that both proteins might favor co-translational import [20].”

Point 18: Minor general comments:

Authors should doublecheck for further lowercase/uppercase, and formatting inconsistencies, e.g., Fig. 1 legend, “24 hours” or “4h” (decision for one format; if second, space missing), Suppl. Figs. S1 figure legend, “D. Wild-Type” (“type” in lowercase), S2B column headers “log2(FC)” (“2” in subscript), “Pvalue” (missing space), S6B, y-axis “Protein Half-life” (lower case “half”); line 71: “…, 3 proteins…” (change to “three protein”), see also lines 138/139 “Only 3 enzymes…” (change to “three enzymes”), or Figure S1: “…of the 3 BioID cell …” (change to “three BioID”), etc.

Lines 59/72/81/243/473: Lowercase/uppercase inconsistency for Drosophila

When p-values are given in the figures, spaces are missing.

Response 18: We modified all the minor general comments and added the required missing information in figures and text as such:

Page 6: “24 hours”; Page 23: “E. Wild type”; Page 24: the “2” of “log2(FC)” was put in subscript in Table S1 column header; Page 24: “p value”; Page 27: Modification of figure S7B y-axis and replacement of “Protein Half-life” with “Protein half-life”; Replacement of all the “3” by “three”, on pages 4-5-6-7-9-10-12-14-16-18-19-20-21-23-24-25-26-27; All the “Drosophila” inconsistencies were corrected, on pages: 4-8-15; All the missing spaces between “p” and “=” were added in the figures 1C (page 6), 3E (page 9), S5C-E (page 26).

References:

  1. Kwak, C.; Shin, S.; Park, J. S.; Jung, M.; My Nhung, T. T.; Kang, M. G.; Lee, C.; Kwon, T. H.; Park, S. K.; Mun, J. Y.; Kim, J. S.; Rhee, H. W. Contact-ID, a Tool for Profiling Organelle Contact Sites, Reveals Regulatory Proteins of Mitochondrial-Associated Membrane Formation. Proc. Natl. Acad. Sci. U. S. A. 2020, 117 (22), 12109–12120. https://doi.org/10.1073/PNAS.1916584117/SUPPL_FILE/PNAS.1916584117.SM01D.AVI.
  2. Cho, K. F.; Branon, T. C.; Rajeev, S.; Svinkina, T.; Udeshi, N. D.; Thoudam, T.; Kwak, C.; Rhee, H. W.; Lee, I. K.; Carr, S. A.; Ting, A. Y. Split-TurboID Enables Contact-Dependent Proximity Labeling in Cells. Proc. Natl. Acad. Sci. U. S. A. 2020, 117 (22), 12143–12154. https://doi.org/10.1073/PNAS.1919528117/SUPPL_FILE/PNAS.1919528117.SD02.XLSX.
  3. Sen, A.; Kalvakuri, S.; Bodmer, R.; Cox, R. T. Clueless, a Protein Required for Mitochondrial Function, Interacts with the PINK1-Parkin Complex in Drosophila. Dis. Model. Mech. 2015, 8 (6), 577–589. https://doi.org/10.1242/dmm.019208.
  4. Sen, A.; Cox, R. T. Clueless Is a Conserved Ribonucleoprotein That Binds the Ribosome at the Mitochondrial Outer Membrane. Biol. Open 2016, 5 (2), 195–203. https://doi.org/10.1242/bio.015313.
  5. Vardi-Oknin, D.; Arava, Y. Characterization of Factors Involved in Localized Translation Near Mitochondria by Ribosome-Proximity Labeling. Front. Cell Dev. Biol. 2019, 7, 305. https://doi.org/10.3389/fcell.2019.00305.
  6. Jan, C. H.; Williams, C. C.; Weissman, J. S. Principles of ER Cotranslational Translocation Revealed by Proximity-Specific Ribosome Profiling. Science 2014, 346 (6210), 1257521. https://doi.org/10.1126/science.1257521.
  7. Pla‐Martín, D.; Schatton, D.; Wiederstein, J. L.; Marx, M.; Khiati, S.; Krüger, M.; Rugarli, E. I. CLUH Granules Coordinate Translation of Mitochondrial Proteins with MTORC1 Signaling and Mitophagy. EMBO J. 2020, 39 (9). https://doi.org/10.15252/embj.2019102731.
  8. Pfanner, N.; Geissler, A. Versatility of the Mitochondrial Protein Import Machinery. Nat. Rev. Mol. Cell Biol. 2001, 2 (5), 339–349. https://doi.org/10.1038/35073006.
  9. Harbauer, A. B.; Zahedi, R. P.; Sickmann, A.; Pfanner, N.; Meisinger, C. The Protein Import Machinery of Mitochondria-a Regulatory Hub in Metabolism, Stress, and Disease. Cell Metab. 2014, 19 (3), 357–372. https://doi.org/10.1016/j.cmet.2014.01.010.
  10. Becker, T.; Song, J.; Pfanner, N. Versatility of Preprotein Transfer from the Cytosol to Mitochondria. Trends in Cell Biology. Elsevier Ltd July 1, 2019, pp 534–548. https://doi.org/10.1016/j.tcb.2019.03.007.
  11. Mukhopadhyay, A.; Ni, L.; Weiner, H. A Co-Translational Model to Explain the in Vivo Import of Proteins into HeLa Cell Mitochondria. Biochem. J. 2004, 382 (1), 385–392. https://doi.org/10.1042/BJ20040065.
  12. Devaux, F.; Lelandais, G.; Garcia, M.; Goussard, S.; Jacq, C. Posttranscriptional Control of Mitochondrial Biogenesis: Spatio-Temporal Regulation of the Protein Import Process. FEBS Lett. 2010, 584 (20), 4273–4279. https://doi.org/10.1016/j.febslet.2010.09.030.
  13. Eliyahu, E.; Pnueli, L.; Melamed, D.; Scherrer, T.; Gerber, A. P.; Pines, O.; Rapaport, D.; Arava, Y. Tom20 Mediates Localization of MRNAs to Mitochondria in a Translation-Dependent Manner. Mol. Cell. Biol. 2010, 30 (1), 284–294. https://doi.org/10.1128/MCB.00651-09.
  14. Liu, Z.; Chen, O.; Wall, J. B. J.; Zheng, M.; Zhou, Y.; Wang, L.; Ruth Vaseghi, H.; Qian, L.; Liu, J. Systematic Comparison of 2A Peptides for Cloning Multi-Genes in a Polycistronic Vector. Sci. Rep. 2017, 7 (1), 1–9. https://doi.org/10.1038/s41598-017-02460-2.
  15. Yogev, O.; Karniely, S.; Pines, O. Translation-Coupled Translocation of Yeast Fumarase into Mitochondria in Vivo. J. Biol. Chem. 2007, 282 (40), 29222–29229. https://doi.org/10.1074/jbc.M704201200.
  16. Gallagher, J. E. G.; Ser, S. L.; Ayers, M. C.; Nassif, C.; Pupo, A. The Polymorphic PolyQ Tail Protein of the Mediator Complex, Med15, Regulates the Variable Response to Diverse Stresses. Int. J. Mol. Sci. 2020, Vol. 21, Page 1894 2020, 21 (5), 1894. https://doi.org/10.3390/IJMS21051894.
  17. Ishikawa, H.; Tachikawa, H.; Miura, Y.; Takahashi, N. TRIM11 Binds to and Destabilizes a Key Component of the Activator-Mediated Cofactor Complex (ARC105) through the Ubiquitin-Proteasome System. FEBS Lett. 2006, 580 (20), 4784–4792. https://doi.org/10.1016/J.FEBSLET.2006.07.066.
  18. Li, J.; Cai, Z.; Vaites, L. P.; Shen, N.; Mitchell, D. C.; Huttlin, E. L.; Paulo, J. A.; Harry, B. L.; Gygi, S. P. Proteome-Wide Mapping of Short-Lived Proteins in Human Cells. Mol. Cell 2021, 81 (22), 4722-4735.e5. https://doi.org/10.1016/J.MOLCEL.2021.09.015.
  19. Mathieson, T.; Franken, H.; Kosinski, J.; Kurzawa, N.; Zinn, N.; Sweetman, G.; Poeckel, D.; Ratnu, V. S.; Schramm, M.; Becher, I.; Steidel, M.; Noh, K. M.; Bergamini, G.; Beck, M.; Bantscheff, M.; Savitski, M. M. Systematic Analysis of Protein Turnover in Primary Cells. Nat. Commun. 2018 91 2018, 9 (1), 1–10. https://doi.org/10.1038/s41467-018-03106-1.
  20. Zhang, Y.; Chen, Y.; Gucek, M.; Xu, H. The Mitochondrial Outer Membrane Protein MDI Promotes Local Protein Synthesis and MtDNA Replication. EMBO J. 2016, 35 (10), 1045–1057. https://doi.org/10.15252/embj.201592994.

Reviewer 2 Report

TOM 20 staining in Fig 1 is punctate or feint and a number of cells show no evidence of any import giving mixed populations within a single panel. These is a little unsettling and should be explained.

Control constructs for the BioID process and methodology are appropriate.

If the turboID is meant to react quickly, why is the incubation with biotin so long (24hours)? The sorter time period may have been reflective of a more physiological situation. If this gave few hits then the reason might be few co-translational chaperones.

Why is nucleic acid binding so low if RNA binding is so high in Fig 2B, as the RNA binding would be expected to be included in the overall nucleic acid binding ?

Although essentially a report of negative results, the data is clearly presented and the manuscript well written.

Colocalisation studies at confocal level are not the most reliable and super-resolution is better. The conclusions must make this point and be a little cautious about over interpretation of the confocal data. If AKAP data had  too high a background and the figure was not shown it is not appropriate to include derived figures for the colocalisation, especially since the graph Fig 4C implies that it has the highest colocalisation with TOM20.

In order for TOM20-TBID to be a reliable marker of intra-mitochondrial protein half-lives, the ability to be sure that the biotinylation of the proteins does not affect the half-life has to be established. There is published evidence (Bogenhagen) to indicate that the half-life of specific proteins varies depending on how quickly they are incorporated into a relevant complex. Thus adding a layer of complexity above the possible change as a consequence of the biotin moiety. It may therefore not be a robust tool for evaluation of this parameter.

Author Response

Response to Reviewer 2 Comments

Dear Editors, Dear Reviewer,

It is our pleasure to re-submit our revised manuscript entitled: “Endogenous TOM20 proximity labeling: a Swiss-knife for the study of mitochondrial proteins in human cells” (ijms-2129480) for consideration as a publication in International Journal of Molecular Sciences.

We addressed the different comments and issues raised, as detailed in the comment below. Thanks to these comments, we believe we managed to clarify the message of our publication, as well as to strengthen the conclusions we initially made.

We thank you very much in advance for re-considering our revised manuscript for publication.

Sincerely Yours,

  1. Meurant and P. Renard.

Point 1: TOM 20 staining in Fig 1 is punctate or feint and a number of cells show no evidence of any import giving mixed populations within a single panel. These is a little unsettling and should be explained.

Response 1: First of all, we would like to thank the reviewer 2 for his precious comments that helped us to substantially improve the manuscript.
The fact that some cells do not show any evidence of reporter protein import is due to the fact those cells are either not transfected or do not express the reporter construct at a detectable level. Indeed, we used and experimental design of a 4 hours transfection period followed by 24 hours of trimethoprim treatment, resulting in only 28 hours total for transgene expression. This short time most likely explains why only few cells are positive for the myc staining since the optimal level of expression is reached after 48 hours. This has been further emphasized in the manuscript as follows:

Page 5: “In the transfected cells expressing the reporters, all constructs display a mitochondrial localization as demonstrated by the colocalization of each construct with TOM20, a protein of the outer mitochondrial membrane (Fig1.A).”

Point 2: If the turboID is meant to react quickly, why is the incubation with biotin so long (24hours)? The sorter time period may have been reflective of a more physiological situation. If this gave few hits then the reason might be few co-translational chaperones.

Response 2: It was indeed unexpected since the initial paper using the miniTurbo variant describes an efficient biotinylation within 10 minutes [1] while, in our hands, a 24 hours-time frame was required to obtain sufficient biotinylation. However, in the initial paper, the protein was overexpressed whereas we have worked with endogenous level of TOM20 expression, which may explain the lower abundance and activity of the biotin ligase. A shorter time period would have indeed been interesting to test and could be achieved by using more recent variants of the biotin ligase, such as ultraID, which shows a strong activity even under physiological expression level and within 10 minutes of incubation with biotin [2]. This has now been further discussed in the manuscript as follows:

Page 15: “Interestingly, more recent variants of the BirA* protein, microID and ultraID, were developed recently and show an even higher labeling efficiency with a 10 minutes biotinylation time showing to be sufficient, even under physiological expression [2].”

Point 3: Why is nucleic acid binding so low if RNA binding is so high in Fig 2B, as the RNA binding would be expected to be included in the overall nucleic acid binding ?

Response 3: This can be explained by the GO terms hierarchy itself. Indeed, the “RNA binding” GO term is one of the members of the “nucleic acid binding” family along with several other members, such as “DNA binding” or “DNA/RNA hybrid binding”. Only the “RNA binding” family of GO term is strongly enriched, in accordance with multiple subfamilies also found among the significantly enriched GO terms such as the “mRNA binding” and “tRNA binding” GO terms (see Fig2.B). The “DNA binding” and “DNA/RNA hybrid binding” GO terms are not enriched in our dataset, which is totally coherent with the localization of the biotin ligase, at the surface of the outer mitochondrial membrane”. This explains why the level of enrichment of the global “Nucleic acid binding” is relatively low.

Point 4: Colocalisation studies at confocal level are not the most reliable and super-resolution is better. The conclusions must make this point and be a little cautious about over interpretation of the confocal data. If AKAP data had too high a background and the figure was not shown it is not appropriate to include derived figures for the colocalisation, especially since the graph Fig 4C implies that it has the highest colocalisation with TOM20.

Response 4: We totally agree with this comment and we revised the conclusion to be more cautious. However, here we used the Airyscan technology which substantially improves the resolution of the confocal microscope and is already closer to super-resolution compared to classical confocal microscopy (120 nm instead of 200 nm of resolution).

Regarding the comment made on the AKAP1 micrograph analysis that should be discarded, we first showed a comparable micrography in Figure 3A for the reader to appreciate the AKAP1 staining. It can be observed that the background is, in fact, not that high and this staining can thus be reasonably used as a “mitochondrially localized positive control”. Actually, we propose an explanation to the moderate level of AKAP1-mtHSP70 colocalization (36 % in Fig.4C): the arbitrary threshold of fluorescence intensity selected for the colocalization analysis was quite stringent, particularly to totally get rid of most of the background noise generated by the different antibodies used in the analyses. This is now further outlined in the manuscript as follows:

Page 10: “Due to the combination of the background noise in the fluorescence signal for AKAP1 and the use of stringent thresholds for fluorescence intensity measurements, arbitrarily set up for the colocalization analysis, the colocalization proportions of AKAP1 with mtHSP70 signal is only of 36 %. In addition, while the colocalization of the three proteins with mtHSP70 was not as strong as for the AKAP1 positive control, it was more important than the colocalization of LARP4 with mtHSP70, and significantly different for MED15 and GPATCH4 (Fig4.C). A similar mitochondrial localization of the three proteins was observed in HEK293T cells (SupFig6.A-C) and in HCT116 cells (SupFig6.D). In conclusion, these colocalization analyses support the mitochondrial localization of the three nuclear proteins suggested by the TOM20-BioID analysis. To further determine the submitochondrial localization of the three proteins, super-resolution microscopy analyses would be required.”

Point 5: In order for TOM20-TBID to be a reliable marker of intra-mitochondrial protein half-lives, the ability to be sure that the biotinylation of the proteins does not affect the half-life has to be established. There is published evidence (Bogenhagen) to indicate that the half-life of specific proteins varies depending on how quickly they are incorporated into a relevant complex. Thus adding a layer of complexity above the possible change as a consequence of the biotin moiety. It may therefore not be a robust tool for evaluation of this parameter.

Response 5: We thank the reviewer 2 for this important remark stressing out the necessity of first checking that the biotinylation of the mitochondrial proteins doesn’t affect their half-life through reducing their assembly potential into larger complexes. We performed a preliminary experiment to get more information on the putative effect of the biotinylation on the half-life of some mitochondrial proteins (Figure X1).

By using cycloheximide, we compared the degradation rate of respiratory complexes proteins (ATP5A, UQCRC2 and NDUFB8) as well as TFB2M protein, reported to be short half-lived in HCT116 [3] and detected in our BioID, in WT HCT116 and in our TOM20-mTb cell lines, in the presence of biotin (FigX1.A). We couldn’t observe any decrease in the half-life of any of the assessed proteins, not even for TFB2M (FigX1.B) which was previously reported to have a half-life of 3 hours in HCT116 cells [3]. This discrepancy might be explained by the fact the observation made for TFB2M was done using a pulse-chase labeling and a proteomic approach rather than using western blot, which is much less accurate. We thus cannot conclude with certainty that the biotinylation doesn’t impact the half-life of mitochondrial proteins since we couldn’t replicate the observations previously done, for TFB2M at least, for an unknown reason. Nonetheless, we don’t observe any reduced abundance or degradation rate of the assessed proteins in the TOM20-mTb cell line compared to WT HCT116. This observation suggests that the biotinylation doesn’t seem to reduce the half-life of those mitochondrial proteins, and more particularly of subunits of respiratory complexes, suggesting a limited effect (if any) on the assembly of those complexes. Of course, additional experiments should be performed to further validate this observation such as pulse-chase SILAC-based analyses of mitochondrial ribosomal proteins or complex I subunits, in the TOM20-mTb cell line, to investigate the assembly of the complexes as previously described [4,5].

These considerations were emphasized in the discussion as follows:

Page 17: “In addition, it is also of importance to validate that the direct biotinylation of mitochondrial proteins has no effect on their half-life time. Indeed, we cannot exclude that the addition of biotin moieties on proteins might impair their ability to assemble into multimolecular complexes, and it has been reported that unassembled subunits of larger complexes are quickly degraded, as demonstrated for mitoribosomes and respiratory chain complexes [4,5]. Therefore, a first step before validating the application of the TOM20-mTb cell line as a predictive tool for mitochondrial protein half-life time would be to validate that the biotin moiety does not affect the potential of proteins to assemble into larger complexes. This could be achieved, as previously described [4,5], using pulse-chase SILAC-based analyses of mitochondrial ribosomal proteins or complex I subunits, in the TOM20-mTb cell line.”

Figure X1. Effect of mitochondrial protein biotinylation on their half-life.   
A.
TOM20-mTb and WT HCT116 cells were seeded and treated with 50 µM biotin and 50 µg/mL of cycloheximide (CHX) (#C7698-1G, Sigma-Aldrich), as described before [3], and were harvested after 0, 3, 6, 12 or 24 hours. The cells were then lysed in RIPA buffer and the same volume of lysate was resolved by SDS-PAGE. ATP5A, UQCRC2 and NDUFB8 were analyzed by western blot using the anti-OXPHOS antibody (#ab110413, Abcam) as well as TFB2M (#24411-1-AP, Proteintech), reported to have a short half-life [3]. The vinculin (#V9131, Sigma-Aldrich) was revealed as a loading control. B. Quantification of the signal for TFB2M protein is normalized to the signal of its respective loading control (vinculin signal) and is then expressed relatively to 0-hour time point. Data are represented as mean ± SD (n = 3). A two-way ANOVA was performed and the p value results for the cell line and time dimensions are displayed.

References:

  1. Branon, T. C.; Bosch, J. A.; Sanchez, A. D.; Udeshi, N. D.; Svinkina, T.; Carr, S. A.; Feldman, J. L.; Perrimon, N.; Ting, A. Y. Efficient Proximity Labeling in Living Cells and Organisms with TurboID. Nature Biotechnology. Nature Publishing Group October 1, 2018, pp 880–898. https://doi.org/10.1038/nbt.4201.
  2. Zhao, X.; Bitsch, S.; Kubitz, L.; Schmitt, K.; Deweid, L.; Roehrig, A.; Barazzone, E. C.; Valerius, O.; Kolmar, H.; Béthune, J. UltraID: A Compact and Efficient Enzyme for Proximity-Dependent Biotinylation in Living Cells. bioRxiv 2021, 2021.06.16.448656. https://doi.org/10.1101/2021.06.16.448656.
  3. Li, J.; Cai, Z.; Vaites, L. P.; Shen, N.; Mitchell, D. C.; Huttlin, E. L.; Paulo, J. A.; Harry, B. L.; Gygi, S. P. Proteome-Wide Mapping of Short-Lived Proteins in Human Cells. Mol. Cell 2021, 81 (22), 4722-4735.e5. https://doi.org/10.1016/J.MOLCEL.2021.09.015.
  4. Bogenhagen, D. F.; Haley, J. D. Pulse-Chase SILAC-Based Analyses Reveal Selective Oversynthesis and Rapid Turnover of Mitochondrial Protein Components of Respiratory Complexes. J. Biol. Chem. 2020, 295 (9), 2544–2554. https://doi.org/10.1074/JBC.RA119.011791.
  5. Bogenhagen, D. F.; Ostermeyer-Fay, A. G.; Haley, J. D.; Garcia-Diaz, M. Kinetics and Mechanism of Mammalian Mitochondrial Ribosome Assembly. Cell Rep. 2018, 22 (7), 1935–1944. https://doi.org/10.1016/J.CELREP.2018.01.066.

Round 2

Reviewer 1 Report

Compared to their initial submission, the authors have considerably improved their manuscript by gaplessly addressing all comments in a highly appreciable, thorough manner. They have enhanced their textual work, including requested references to other publications, provided additional experimental data where feasible, and provided a clear argumentation and comprehensible responses. Also, I want to thank the authors for sharing the access information to the yet not officially available mass spectrometry data. Overall, this solid revision has reached the desired scientific quality, and I recommend the manuscript for publication.   

Two minor points:

  1. I might have missed it, but I could not find the revised graphic abstract in the manuscript file.
  2. Line 224: change “didn’t” to “did not”

Author Response

Response to Reviewer 1 Comments

Dear Editors, Dear Reviewer,

We were really glad to read the positive comments of Reviewer 1 regarding the revised version of our manuscript entitled: “Endogenous TOM20 proximity labeling: a Swiss-knife for the study of mitochondrial proteins in human cells” (ijms-2129480) and we would like to take this opportunity to thank once again Reviewer 1 for his constructive comments and suggestions.

We addressed the different remaining comments and issues raised by the reviewers, as detailed below.

We thank you very much in advance for considering our revised manuscript for publication.

Sincerely Yours,

  1. Meurant and P. Renard.

Point 1: Compared to their initial submission, the authors have considerably improved their manuscript by gaplessly addressing all comments in a highly appreciable, thorough manner. They have enhanced their textual work, including requested references to other publications, provided additional experimental data where feasible, and provided a clear argumentation and comprehensible responses. Also, I want to thank the authors for sharing the access information to the yet not officially available mass spectrometry data. Overall, this solid revision has reached the desired scientific quality, and I recommend the manuscript for publication.   

Two minor points:

  1. I might have missed it, but I could not find the revised graphic abstract in the manuscript file.
  2. Line 224: change “didn’t” to “did not”

Response 1:

In response to the first minor point, the graphical abstract was modified as demanded and resubmitted in the first round of revision. However, apparently the graphical abstract is not included in the formatted version of the manuscript, so we join here below the last version of the graphical abstract:

For the second point, we modified accordingly the “didn’t” into “did not” at the line 224.

Reviewer 2 Report

The quality of the TOM20 IF is not of high enough quality.

The number of repeats for the co-localisation (Fig 1, Fig 3, Fig 4) is inadequate to make the conclusions given.

RNA-binding proteins score highly in fig 2 but are the lowest in Fig 5. Is this consistent ?

Author Response

Response to Reviewer 2 Comments

Dear Editors, Dear Reviewer,

It is our pleasure to re-submit our revised manuscript (round 2) entitled: “Endogenous TOM20 proximity labeling: a Swiss-knife for the study of mitochondrial proteins in human cells” (ijms-2129480) for consideration as a publication in International Journal of Molecular Sciences.

We addressed the different remaining comments and issues raised by the reviewers, as detailed below.

We thank you very much in advance for considering our revised manuscript for publication.

Sincerely Yours,

  1. Meurant and P. Renard.

Point 1: The quality of the TOM20 IF is not of high enough quality.

Response 1: We tried to improve as best as we could the micrographs for TOM20. In the Figure 1B, we therefore changed the micrograph for preARG2-DHFR and we also improved the signal for the preOTC-DHFR to better visualize the TOM20 staining. However, in all the trimethoprim conditions, we cannot really improve the TOM20 staining, which is indeed of a lower quality. Indeed, we noticed that the trimethoprim treatment seems to alter the morphology and probably has an impact on the TOM20 marker resulting in a reduced quality of mitochondrial staining. In addition, the HCT116 and HEK293T cells are small cells with a high nucleus/cytoplasm ratio and the mitochondrial network in those cells is already mostly fractionated which altogether contribute to a suboptimal mitochondrial staining compared to other cell lines.

Here below you can find the modified Figure 1 to appreciate the modifications:

Point 2: The number of repeats for the co-localisation (Fig 1, Fig 3, Fig 4) is inadequate to make the conclusions given.

Response 2: We do not agree with this comment, for the reasons explained below.

First of all, for the Figures 1 and 3, the purpose of the quantifications rather aimed at supporting a mitochondrial or cytosolic subcellular localization than really assessing a colocalization between two proteins. Indeed, the different reporters are devoted to assess mitochondrial protein import (co-translationally or post-translationally) with either a cytosolic or mitochondrial localization of the different reporters. When the import pathway of the reporter protein is blocked (in response to trimethoprim treatment), the reporter completely and clearly localizes in the cytosol, the only exception being preCOX4I1 that we did not select for the co-translational import assessment. Therefore, the result can already be clearly appreciated by visualizing the cells in the micrographs presented in these figures. The quantification is thus not aimed at providing specific co-localization values but only for providing graphical representation of the clearcut localization being either mitochondrial or cytosolic.
For the quantifications made in the Figure 4, an average of 60-70 cells were counted in total in three independent biological replicates, for both NHDFs and HEK293T cells and the result is consistent in the different cell lines that have been analyzed. Moreover, a such number of cells for confocal co-localization analyses is commonly accepted in high impact-factor journals (Wan et al, EMBO J. 2023, e112387) [1], including in IJMS (Li et al, Int. J. Mol. Sci. 2023, 24 (2), 1710; Chu et al, Int. J. Mol. Sci. 2023, 24 (5), 4867) [2,3]. Finally, the variability between the different replicates is quite low, suggesting that the number of cells analyzed in our study is sufficient to highlight significant differences.

Point 3: RNA-binding proteins score highly in fig 2 but are the lowest in Fig 5. Is this consistent ?

Response 3: We are not sure to understand this comment since the “RNA binding” Gene Ontology (GO) term is not found in the Figure 5. Indeed, for the DAVID-based GO enrichment analysis shown in the Figure 5A, only the Cell Component family of GO terms is displayed and it is not possible to find the “RNA binding” term, part of the Molecular Function family. Similarly, the density plot shown in Figure 5E displays GO terms of the Cell Component family. For the Figure 5D, the emapplot displays GO terms of the three families of GO terms but does not display the “RNA binding” term. This is not surprising, considering the principle of the emapplot which aims at showing the strongest links between the most enriched GO terms found in the gene set enrichment analysis [4]. The most enriched GO terms emerging from our BioID dataset point toward the high enrichment of mitochondria-related GO terms, supporting that the TOM20-mTb cell line can be used to detect the entry of mitochondrial proteins. It is thus not surprising that the “RNA binding” term, which is not specifically connected to mitochondria-related GO terms, could not be found in the emapplot.

References:

  1. Wan, W.; Qian, C.; Wang, Q.; Li, J.; Zhang, H.; Wang, L.; Pu, M.; Huang, Y.; He, Z.; Zhou, T.; Shen, H.-M.; Liu, W. STING Directly Recruits WIPI2 for Autophagosome Formation during STING-Induced Autophagy. EMBO J. 2023, e112387. https://doi.org/10.15252/EMBJ.2022112387.
  2. Li, J.; Chen, H.; Cai, L.; Guo, D.; Zhang, D.; Zhou, X.; Xie, J. SDF-1α Promotes Chondrocyte Autophagy through CXCR4/MTOR Signaling Axis. Int. J. Mol. Sci. 2023, 24 (2), 1710. https://doi.org/10.3390/IJMS24021710/S1.
  3. Chu, A.; Yao, Y.; Saffi, G. T.; Chung, J. H.; Botelho, R. J.; Glibowicka, M.; Deber, C. M.; Manolson, M. F. Characterization of a PIP Binding Site in the N-Terminal Domain of V-ATPase A4 and Its Role in Plasma Membrane Association. Int. J. Mol. Sci. 2023, Vol. 24, Page 4867 2023, 24 (5), 4867. https://doi.org/10.3390/IJMS24054867.
  4. Wu, T.; Hu, E.; Xu, S.; Chen, M.; Guo, P.; Dai, Z.; Feng, T.; Zhou, L.; Tang, W.; Zhan, L.; Fu, X.; Liu, S.; Bo, X.; Yu, G. ClusterProfiler 4.0: A Universal Enrichment Tool for Interpreting Omics Data. Innov. 2021, 2 (3). https://doi.org/10.1016/J.XINN.2021.100141.

Round 3

Reviewer 2 Report

I am not convinced this is sufficient rigour. The response to the comment about localisation still holds. 4 points on a graph is too few for this type of analysis. Also the text still makes statements such as “/Even upon trimethoprim treatment the encoded re-298 porter protein still localized inside the mitochondria in LARP4 KO cells//(*SupFig5.E*) /”.  I can’t find new Supp figs and the old SF5 didn’t have a panel E. The resolution on the images is not sufficient to distinguish inside from associated but on the outside.